



# Ionosonde and GPS Total Electron Content Observations during the 26 December 2019 Annular Solar Eclipse over Indonesia

Jiyo Harjosuwito[1], Asnawi Husin[1], Varuliantor Dear[1], Johan Muhamad[1], Agri Faturahman[1],
Afrizal Bahar[2], Erlansyah Erlansyah[3], Agung Syetiawan[4], and Rezy Pradipta[5]

[1]Research Center for Space, Research Organization for Aeronautics and Space (LAPAN), National Research and Innovation Agency (BRIN), Indonesia
[2]Agam Atmospheric and Space Observation Office, Research Organization for Aeronautics and Space (LAPAN), National Research and Innovation Agency (BRIN), Indonesia
[3]Pontianak Atmospheric and Space Observation Office, Research Organization for Aeronautics and Space (LAPAN), National Research and Innovation Agency (BRIN), Indonesia
[4]Research Center for Geospatial, Research Organization for Earth Sciences and Maritime, National Research and Innovation Agency (BRIN), Indonesia
[5]Institute for Scientific Research, Boston College, United States

**Correspondence:** Rezy Pradipta (rezy.pradipta@bc.edu)

**Abstract.** We report our investigation of ionospheric effects due to the passage of an annular solar eclipse over Southeast Asia on 26 December 2019, using multiple set of observations. Two ionosondes (one at Kototabang and another at Pontianak) were used to measure dynamical changes in the ionospheric layer during the event. A network of ground-based GPS receiver stations in Indonesia were used to derive the distribution of total electron content (TEC) over the region. In addition, extreme

ultraviolet (EUV) images of the Sun from the Atmospheric Imaging Assembly (AIA) instrument on board the Solar Dynamics Observatory (SDO) satellite were also analyzed to determine possible impacts of solar active regions on the changes that occurred in the ionosphere during the eclipse. We found $-1.67$ MHz and $-1.58$ MHz reduction (23.2% and 22.4% relative reduction) in foF2 during the solar eclipse over Kototabang and Pontianak, respectively. The respective TEC reduction over Kototabang and Pontianak during the eclipse was $-4.34$ TECU and $-5.45$ TECU (24.9% and 27.9% relative reduction).

Overall, there was 34–36 minutes delay from maximum eclipse until minimum foF2 was reached at these two locations. The corresponding time delays for eclipse-related TEC reduction at these two locations were 40 minutes and 16 minutes, respectively. The ionospheric F-layer was found to descend with a speed of 9–19 m/s during the first half of the eclipse period. We also found an apparent rise of the ionospheric F-layer height near the end of the solar eclipse period, equivalent to vertical drift velocity of 44–47 m/s. The GPS TEC data mapping along a set of cross-sectional cut lines indicate that the greatest

TEC reduction actually occurred to the north of the solar eclipse path, opposite of the direction from which the lunar shadow fell. As the central path of the solar eclipse was located just to the north of the southern equatorial ionization anomaly (EIA) crest, it is suspected that such a peculiar TEC reduction pattern was caused by plasma flow associated with the equatorial fountain effect. Net perturbations of TEC were also computed and analyzed, which revealed the presence some wavelike fluctuations associated with the solar eclipse event. Some of the observed TEC perturbation patterns that propagated with a

velocity matching the lunar shadow may be explained in terms of non-uniform EUV illumination that arose as various active



regions on the Sun went obstructed and unobstructed during the eclipse. The remaining wavelike features are likely to be traveling ionospheric disturbances (TIDs) driven by acoustic-gravity waves (AGWs), generated by the passage of the solar eclipse on top of other diurnal factors.

# 1 Introduction

On 26 December 2019, an annular solar eclipse occurred over the Indonesian sector, specifically over the Sumatra and Kalimantan (Borneo) islands. As an astronomical phenomenon that occurs only a few times a year over varying areas of the globe, any given solar eclipse event is a unique historical marker that is valuable to investigate from a scientific standpoint. An interesting aspect of solar eclipse events is their effects on the Earth's atmosphere, including the upper atmosphere or the ionosphere. Research works on the effects of solar eclipses on the Earth's atmosphere had started as early as the 1830s (e.g. Birt, 1836; Aplin et al., 2016 and references therein). The effects of solar eclipses on electromagnetic aspects of the ionosphere had been studied since the early 20th century (see e.g. Aplin et al., 2016 for an overview). Likewise, studies on the effects of solar eclipses on physical and chemical processes in the ionosphere had begun around a century ago (Mitra, 1933; Beynon, 1955; Ratcliffe, 1956; Rishbeth, 1968 and references therein). Observable effects that solar eclipses might produce on the ionosphere had been quite extensively documented in the peer-reviewed scientific literature (Burton and Boardman, 1933; Cheng et al., 1992; Bamford, 2001; Mansoori et al., 2011; Adekoya et al., 2015). However, any particular solar eclipse event occurs under its own unique set of background conditions, including geographical location and solar local time. Different solar eclipse events may produce varying effects on the ionosphere depending whether they occur near dawn, around noon, or near dusk (Setty, 1960; Sridharan et al., 2002; St.-Maurice et al., 2011; Chen et al., 2013; Chuo, 2013; Adekoya et al., 2019). Solar eclipse events that occur over the low-latitude, midlatitude, and high-latitude regions may also produce different ionospheric effects (Kurkin et al., 2001; Jakowski et al., 2008; Le et al., 2009; Chen et al., 2011; Kumar et al., 2013; Chukwuma and Adekoya, 2016). Likewise, solar eclipse events that occur during different phases of the 11-year solar cycle may produce different ionospheric effects as well (Adekoya and Chukwuma, 2012).

Initially, scientific investigations on this subject only relied on basic radio observations of the ionosphere (Evans, 1965; Anastassiades, 1970). Nevertheless, data collected from those basic radio observations have yielded important scientific advances (Gledhill, 1959; Price, 1959; Lerfald et al., 1965; Hanuise et al., 1982). For many decades, research on this subject had continued through the use of various observing equipment and methods, including ground-based observations (e.g. Choudhary et al., 2011; Atulkar et al., 2015; Frissell et al., 2018; Goncharenko et al., 2018; Liu et al., 2019; Jose et al., 2020) and observations using satellites (e.g. Wang et al., 2010; Cherniak and Zakharenkova, 2018; Hairston et al., 2018). Over time, there are growing numbers and varying types of ionospheric observation instruments with a wider spatial coverage and geographical distribution. The use of incoherent scatter radars started to become more common since the 1960s (Gordon, 1964; Perkins et al., 1965). Growing network of ground-based global navigation satellite system (GNSS) receivers for total electron content (TEC) measurements became ubiquitous at the turn of the 21st century (Coster et al., 1992; Afraimovich et al., 2002; Rideout and Coster, 2006; Coster et al., 2017; Zhang et al., 2017; Cherniak and Zakharenkova, 2018). Satellite-borne and ground-based





optical imaging instruments had also been used for solar eclipse observations (Eastes et al., 2008; Harding et al., 2018; Aryal
et al., 2020; Paulino et al., 2020).

There are still very few well-documented studies on the ionospheric effects of solar eclipse events over the Indonesian
sector (Perwitasari and Muslim, 2009; Muslim et al., 2016; Dear and Yulianto, 2016; Vita et al., 2017; Dear et al., 2020).
This deficiency is consistent with the shared notion that the Southeast Asian sector is still a relatively understudied area in
terms of comprehensively analyzed ionospheric observation data, despite the inherently global and collaborative nature of
space research. This regional void have persisted for a long time in part due to limited space science knowledge and lack of
ionospheric observation coverage. As such, a comprehensive examination of various ionospheric effects caused by solar eclipse
events, including that on 26 December 2019, is needed — not only scientifically but also institutionally in order to garner a
greater level of interest in the region. From scientific perspective, it is part of ongoing collective efforts from previous research
works in this particular subject (Anggarani, 2016; Dear et al., 2020; Faturahman et al., 2022a). From institutional perspective,
it is a timely opportunity to push for a more intensive utilization of existing ionospheric observation instruments in the region
as inclusive part of global space science community.

In this paper, we investigated the ionospheric response to the 26 December 2019 annular solar eclipse event over the Indone-
sian sector using a combination of ionosonde and GNSS receiver observations, in conjunction with solar image analysis. The
present work adds to existing analyses of this particular event performed independently by Aa et al. (2020) and Barad et al.
(2022) over the Indian and Southeast Asian longitudes. Section 2 of this paper describes the methodology, Section 3 describes
the observation results, Section 4 presents a discussion of the findings, and Section 5 presents the conclusion.

## 2 Instruments and Methodology

Relatively recently, there have been some notable changes in terms of ionospheric observation instruments within the Indone-
sian sector. In the last two decades or so, network of ground-based GNSS receiver stations in the region have been growing
steadily (Masykur, 2021), which may provide adequate TEC data coverage. In Indonesia, those GNSS receiver stations are part
of larger infrastructure managed by the Geospatial Information Agency (Badan Informasi Geospasial – BIG), the primary pur-
pose of which is geodetic surveying. These GNSS receivers complement the existing ionosondes, which had been the dominant
type of ionospheric observation instruments in Southeast Asia since the 1950s (Sen, 1949; Osborne, 1951; Minnis, 1957) and
in Indonesia since the 1980s (Haggard, 1985, 1988; Gage et al., 1989; Lynn et al., 2000; Lynn et al., 2004). In addition, there
is also the Equatorial Atmosphere Radar (EAR) facility in Kototabang, West Sumatra, which is a collaboration between the
Indonesian Space Agency (LAPAN) and Kyoto University, Japan (Fukao et al., 2003). The current coverage of ionospheric ob-
servation instruments in Indonesia is still incomplete, especially in the eastern territory where observation instruments remain
relatively rare. Perhaps it will take another decade until a more complete observation coverage is finally realized. Nevertheless,
the distribution of existing instruments is more than sufficient to support many ionospheric research activities, including our
examination of the 26 December 2019 annular solar eclipse event (present paper). Figure 1 shows the spatial distribution of
available instruments during the solar eclipse event, in relation to the main eclipse trajectory.



This study used observation data from ionosondes and GNSS receiver network in the Indonesian sector. A separate analysis of the EAR Kototabang observation data during this solar eclipse event is published elsewhere (Faturahman et al., 2022b). In addition, analysis of masked solar images was also performed to quantify the degree of inhomogeneity in the solar EUV

illumination during the eclipse.

## 2.1  Ionosonde Measurements

The ionosondes used for this study are located in Kototabang, West Sumatra (0.20°S 100.32°E) and Pontianak, West Kalimantan (0.02°S 109.33°E). For the 26 December 2019 annular solar eclipse, the maximum solar obscuration at these two locations (cf. Figure 1) ranges between 91%–94%. Since one day prior to the eclipse event until one day after the eclipse event, ionoson-

des at these two stations were set to record one ionogram every 5 minutes. On other days, ionograms were recorded once every 15 minutes.

At the Kototabang station, a frequency-modulated continuous wave (FMCW) ionosonde owned by the Japanese National Institute of Information and Communications Technology (NICT) was in operation. Meanwhile at the Pontianak station, a Canadian Advanced Digital Ionosonde (CADI) owned by the Indonesian National Institute of Aeronautics and Space (LAPAN)

was in operation. Both ionosondes have been in continuous operation since 2004, and they were in the midst of some repair and maintainance work during the 26 December 2019 annular solar eclipse event. As a result, noise level in the recorded ionograms was high, which necessitated manual scaling process for the ionogram data analysis and interpretation. Figure 2 shows sample ionograms that were recorded at these two stations during the solar eclipse event.

Manual scaling of ionospheric parameters from the ionograms was performed based on standard procedures provided in the

UAG-23A handbook (Piggott and Rawer, 1972). Essential ionospheric parameters that were derived through ionogram scaling include the critical frequency of F1-layer (foF1), critical frequency of F2-layer (foF2), and the apparent parabolic layer peak height (hpF2). The scaling process was performed for all ionograms recorded in December 2019. In addition, output from the International Reference Ionosphere (IRI) model was also used to help derive baseline curves for these ionospheric parameters.

## 2.2  Ground-Based GPS TEC Measurements

We derived the regional total electron content (TEC) using global navigation satellite system (GNSS) observation data in receiver-independent exchange (RINEX) format from the Indonesian Continuously Operating Reference Station (INACORS) network, which is maintained by the Indonesian Geospatial Information Agency (BIG). The observation data are catalogued at **http://inacors.big.go.id/sbc/** by the BIG. In 2019, the INACORS network consisted of 207 receiver stations. In this study, we used data from 46 receiver stations that are distributed in Sumatra and Kalimantan (cf. Figure 1). Geographic coordinates of

the INACORS stations that we used in this study are provided in the supplementary material. The number of GNSS receiver stations in Sumatra is greater than that in Kalimantan, since the INACORS network is used primarily for geodetic mapping and monitoring of the Eurasian tectonic plate.

The TEC values were calculated from the receiver-independent exchange (RINEX) observation files using the Gopi data analysis software (Rama Rao et al., 2006; Seemala and Valladares, 2011). Only signals from the Global Positioning System



(GPS) constellation were used in the TEC computation. For the purpose of spatial mapping of TEC data in this study, we selected an altitude of 350 km for the ionospheric piercing points (IPP). Figure 3 shows sample snapshots of TEC observations obtained from basic processing of the data, illustrating the spatial coverage provided by these receiver stations. Observations at 05:30 UTC on 25, 26, and 27 December 2019 are shown in these snapshots; with the IPPs represented as open circles and TEC values indicated using a colormap. The solar eclipse path on 26 December 2019 is shown as dashed blue curve, and the

instantaneous position of the lunar shadow at that time is indicated by solid gray circle. Significant decrease in TEC values around the eclipse path on 26 December 2019 is quite visible in the data map. An animated version of Figure 3 is also provided in the supplementary material.

Further in the analysis, TEC data detrending was also performed. Two types of data detrending were performed: one to derive ΔTEC (general deviations from normal condition) and another to derive TECP (wavelike perturbations with much smaller

amplitudes and finer structures). The derivation of ΔTEC was performed on TEC values that had been spatially mapped from individual IPPs onto fixed grid point(s). The baseline for detrending the TEC data into ΔTEC was based on the average diurnal TEC variation from calendar dates other than the solar eclipse day. In contrast, the derivation of TECP was performed using TEC time series on the individual IPPs. Only after completing the detrending process on the IPPs did we spatially map the TECP values onto fixed grid point(s) for data display. The baseline for detrending the TEC data into TECP was obtained via

30-minute running average of the TEC time series being examined.

## 2.3   Solar EUV Illumination Calculation

We investigated changes in extreme ultraviolet (EUV) radiation that illuminated various points in the Earth's upper atmosphere during the eclipse. For computational purposes, the solar EUV illumination was considered at 100 km altitude above the Earth's surface. EUV images of the Sun were obtained from the Atmospheric Imaging Assembly (AIA) instrument onboard the Solar

Dynamics Observatory (SDO) satellite to represent the solar EUV emissions from the corona. We used an AIA 193 Å image taken at 03:10:04 UTC on 26 December 2019 as the representative solar disk image. Because there was no solar flare activity recorded during the eclipse, we assumed that there was no significant change on the Sun's surface, and therefore a single sample image was sufficient to represent the Sun at all times during the eclipse event. Although there was no solar flare at that time, there were two active regions (ARs) on the Sun with higher levels of emission: one in the southeast of the solar disk and

another one in the northwest of the solar disk.

We have implemented a computer model of solar eclipse that can reproduce the obscuration of the Sun as viewed from any point on Earth. The EUV image of the Sun was placed at the center of the field-of-view of the simulation, and a model of the Moon was moved across the Sun based on the DE421 ephemeris data developed by the Jet Propulsion Laboratory, NASA. We used the Skyfield library in Python programming language to develop the simulation and plotted the sequential motion of the

artificial Moon with one minute time cadence to represent the eclipse. Changes in the EUV radiation received on Earth during the eclipse can be approximated as the change of total intensities of all pixels in the image, with some of the pixels blocked by the artificial Moon. This is based on an approach previously performed by Huba and Drob (2017).





Figure 4 illustrates this modeling of solar EUV illumination during the eclipse. Here we calculated the time evolution of solar illuminance and EUV 193 Å irradiance received at the lower ionosphere (at 100 km altitude) over a geographic site
near the greatest eclipse point in Siak (1.01°N 102.25°E), Indonesia. In order to enhance the eclipse effect on the solar EUV radiation received on Earth, we changed the contrast of the image. This was done to emphasize the presence of ARs on the Sun by darkening the low intensity regions and intensifying the bright regions, such as the ARs. An emphasis on ARs is needed because ARs are small concentrated areas with high brightness, which would abruptly disappear as the solar obscuration gradually changes. By performing this procedure, variation in the soalr EUV radiation (i.e. the EUV obscuration factor) during
the eclipse can become more obvious, as shown by the purple line in the time series plot.

Further, at various epochs during the eclipse, we also computed the total solar EUV irradiance over a number of evenly spaced grid points within a rectangular region from latitude 12°S to 7°N and from longitude 90°E to 150°E with 1 degree spatial resolution. At any given epoch, spatial non-uniformity of the solar EUV irradiance can then be quantified. We followed the approach by Huba and Drob (2017) and Mrak et al. (2018) to identify this non-uniformity by calculating the second spatial
derivative (i.e. the Laplacian) of the distribution of solar EUV radiation received on Earth. This is a reasonable approach, as the Laplacian has been frequently used in image processing for edge detection (e.g. Wang, 2007).

## 3 Observation Results

In this section, we present the results that were obtained from the ground-based radio observation instruments (ionosondes and GNSS receivers), as well as those from the solar EUV illumination analysis.

### 3.1 Ionosonde Observations

Results from ionosonde obervations and manual scalings of the recorded ionograms are presented here in terms of foF2, foF1, and hpF2 variations. In order to provide additional contexts, the corresponding relations to NmF2, NmF1, and apparent vertical drift velocity are also presented.

Figure 5 shows the time series plots of foF2 over Kototabang and Pontianak on 26 December 2019. Based on the ionosonde
observations, changes in foF2 as a response to the solar eclipse were generally found to exhibit two distinct phases: a reduction phase and a recovery phase.

Over Kototabang, reduction in foF2 was seen starting at 04:35 UTC (11:35 LT) from an initial value of 6.70 MHz (NmF2 $\approx 5.54 \times 10^5$ el cm$^{-3}$) to reach its minimum value of 5.50 MHz (NmF2 $\approx 3.73 \times 10^5$ el cm$^{-3}$) at 05:45 UTC (12:45 LT). During the 70 minutes of this reduction phase, foF2 dropped by 1.20 MHz. This foF2 reduction is equivalent to a drop in
NmF2 by $1.81 \times 10^5$ el cm$^{-3}$, which gives us an average rate of ionospheric density reduction of $-43$ el cm$^{-3}$/s. The recovery phase started at 05:45 UTC (12:45 LT) and ended at 08:45 UTC (15:45 LT) with an increase in foF2 by 1.60 MHz, from 5.50 MHz to finally reach 7.10 MHz (NmF2 $\approx 6.22 \times 10^5$ el cm$^{-3}$) over 180 minutes duration. This is equivalent to a rise in NmF2 by $2.49 \times 10^5$ el cm$^{-3}$ during the recovery phase, which gives us an average rate of ionospheric density increase of $+23$ el cm$^{-3}$/s.





Meanwhile over Pontianak, reduction in foF2 started at 04:50 UTC (11:50 LT) from an initial value of 6.21 MHz (NmF2 $\approx 4.76 \times 10^5$ el cm$^{-3}$) to reach its minimum value of 5.44 MHz (NmF2 $\approx 3.65 \times 10^5$ el cm$^{-3}$) at 06:20 UTC (13:20 LT). During the 90 minutes of this reduction phase, foF2 dropped by 0.77 MHz. This foF2 reduction is equivalent to a drop in NmF2 by $1.11 \times 10^5$ el cm$^{-3}$, which gives us an average rate of ionospheric density reduction of $-20.6$ el cm$^{-3}$/s. The recovery phase occurred over a duration of 155 minutes, starting at 06:20 UTC (13:20 LT) until 08:55 UTC (15:55 LT) with an increase in

foF2 by 1.23 MHz (from 5.44 MHz to finally reach 6.67 MHz). This is equivalent to a rise in NmF2 by $1.84 \times 10^5$ el cm$^{-3}$ during the recovery phase, which gives us an average rate of ionospheric density recovery of $+19.9$ el cm$^{-3}$/s.

Observations over Kototabang and Pontianak indicate that there was some lag between the start of the ecipse and the initial decrease in foF2. Over Kototabang, there was a time lag of 77 minutes before foF2 started to decrease, while the corresponding time lag over Pontianak was around 65 minutes. Likewise, the recovery phase extended well beyond the end of the eclipse.

The extra time between the end of the eclipse and the end of recovery phase over Kototabang was 97 minutes, while that over Pontianak was 83 minutes.

With respect to the normal baseline (derived based on an assimilation of the entire December 2019 observation data and IRI model output), the magnitude of foF2 reduction over Kototabang and Pontianak was quite similar. The reduction of foF2 over Kototabang with respect to the normal baseline was $-1.67$ MHz (a 23.2% relative reduction), meanwhile that over Pontianak

was $-1.58$ MHz (a 22.4% relative reduction).

Figure 6 shows the time series plots of foF1 over Kototabang and Pontianak on 26 December 2019. Similar to what we have observed for foF2, changes in the critical frequency of ionospheric F1 layer as a response to the solar eclipse also exhibited two distinct phases: a reduction phase and a recovery phase.

Over Kototabang, reduction in foF1 started at 03:30 UTC (10:30 LT) which was 12 minutes after the start of the eclipse,

with an initial foF1 of 4.13 MHz which corresponds to NmF1 of $2.11 \times 10^5$ el cm$^{-3}$. The minimum foF1 of 2.9 MHz (NmF1 $\approx 1.04 \times 10^5$ el cm$^{-3}$) was reached at 05:20 UTC (12:20 LT) which was 9 minutes after the maximum eclipse. Afterwards, foF1 started its recovery phase to finally reach 4.73 MHz (NmF1 $\approx 2.76 \times 10^5$ el cm$^{-3}$) at 06:25 UTC (13:25 LT) which was 43 minutes before the end of the eclipse.

The duration of the reduction phase was 110 minutes, in which foF1 dropped by 1.23 MHz (from 4.13 MHz to 2.90 MHz).

This foF1 reduction corresponds to a drop in ionospheric density by $1.07 \times 10^5$ el cm$^{-3}$ (from $2.11 \times 10^5$ el cm$^{-3}$ to $1.04 \times 10^5$ el cm$^{-3}$). This gives an average rate of ionospheric density reduction of $-16$ el cm$^{-3}$/s. With respect to the normal baseline value of 4.34 MHz, the minimum foF1 was 1.44 MHz lower (a 33% relative reduction). The recovery phase occurred over a duration of 65 minutes, with an increase in foF1 by 1.83 MHz ($\Delta$NmF1 $\approx 1.72 \times 10^5$ el cm$^{-3}$), which gives us an average rate of ionospheric density recovery of $+44$ el cm$^{-3}$/s.

Meanwhile over Pontianak, observations indicate that foF1 started to decrease at 04:30 UTC (11:30 LT) which was 45 minutes after the start of the eclipse, with an initial foF1 of 4.89 MHz (NmF1 $\approx 2.97 \times 10^5$ el cm$^{-3}$). The minimum foF1 of 3.34 MHz (NmF1 $\approx 1.38 \times 10^5$ el cm$^{-3}$) was reached at 06:00 UTC (13:00 LT) which was 16 minutes after the maximum eclipse. Afterwards, foF1 started its recovery phase to finally reach 4.77 MHz (NmF1 $\approx 2.83 \times 10^5$ el cm$^{-3}$) at 07:20 UTC (14:20 LT) which was 12 minutes before the end of the eclipse.





The foF1 reduction phase over Pontianak took place over 90 minutes duration, in which foF1 dropped by 1.55 MHz (from 4.89 MHz to 3.34 MHz). This foF1 reduction corresponds to a drop in NmF1 by $1.59 \times 10^5$ el cm$^{-3}$ (from $2.97 \times 10^5$ el cm$^{-3}$ to $1.38 \times 10^5$ el cm$^{-3}$). This gives an average rate of ionospheric density reduction of $-29$ el cm$^{-3}$/s. With respect to the normal baseline value of 4.26 MHz, the minimum foF1 was 0.92 MHz lower (a 21% relative reduction). The recovery phase occurred over a duration of 80 minutes, with an increase in foF1 by 1.43 MHz (from 3.34 MHz to 4.47 MHz). This rise in foF1 corresponds to an increase in NmF1 by $1.43 \times 10^5$ el cm$^{-3}$ (from $1.38 \times 10^5$ el cm$^{-3}$ to $2.83 \times 10^5$ el cm$^{-3}$). This gives us an average rate of ionospheric density recovery of $+30$ el cm$^{-3}$/s.

Figure 7 shows the time series plots of hpF2 measurements at Kototabang and Pontianak on 26 December 2019. Based on these observations, changes in hpF2 as a response to the solar eclipse can be categorized into three phases: a descending phase, a rising phase, and a recovery phase.

The ionospheric response over Kototabang in terms of hpF2 is shown in Figure 7a. During the first phase that started at 03:20 UTC (10:20 LT) which was 12 minutes after the start of the eclipse, hpF2 descended from an initial altitude of 516 km. The descent occurred for 95 minutes duration until 04:55 UTC (11:55 LT) when hpF2 reached an altitude of 466 km. With hpF2 descending by 50 km, the average rate of descent during this first phase was approximately $-32$ km/h or $-8.8$ m/s. At 05:30 UTC (12:30 LT), hpF2 started to rise from an altitude of 486 km to reach 682 km at 06:40 UTC (13:40 LT). With hpF2 rising by 196 km during this second phase, the average rate of climb was 168 km/h or 47 m/s. The recovery phase subsequently occurred over 20 minutes duration, ending at 07:00 UTC (14:00 LT) when hpF2 returned to baseline level.

The ionospheric response over Pontianak in terms of hpF2 is shown in Figure 7b. The first phase started at 03:55 UTC (10:55 LT) which was 10 minutes after the start of the eclipse, with an initial hpF2 of 616 km. This phase occurred over 120 minutes duration until 05:55 UTC (12:55 LT) when hpF2 reached an altitude of 480 km. In other words, hpF2 descended by 136 km, which means that the average rate of descent was $-68$ km/h or $-19$ m/s. The second phase occurred for 100 minutes duration, starting from 05:55 UTC (12:55 LT) until 07:35 UTC (14:35 LT), during which hpF2 rose from 480 km to 747 km. With hpF2 rising by 267 km, the average rate of climb was 160 km/h or 44 m/s. The recovery phase subsequently occurred over 15 minutes duration, ending at 07:40 UTC (14:40 LT) when hpF2 returned to baseline level.

## 3.2 GPS TEC Observations

Figure 8 shows a map depicting the location of 4 discrete checkpoints where we examined the TEC time series data. These checkpoints are code-named KTB (co-located with Kototabang station), GEC (point of greatest eclipse), GDU (point of longest annularity duration), and PTK (co-located with Pontianak station). By examining the TEC time series at these 4 fixed check-points, we remove some of the ambiguity that may arise from the movement of the IPPs. The TEC values from individual IPPs are spatially interpolated onto the checkpoints using a form of inverse-distance weighting (IDW) interpolation technique (Pradipta et al., 2014). The westernmost and easternmost checkpoints (KTB and PTK) were chosen since there are ionosondes at these locations. The other checkpoints (GEC and GDU) were chosen because of the special condition of the solar eclipse parameters there. At each checkpoint, time series of absolute TEC and ∆TEC were examined. The ∆TEC values were obtained by subtracting a smooth baseline from the absolute TEC values, which would reveal net changes in TEC relative to normal





condition. The baseline TEC was determined by averaging the TEC values from 25 and 27 December 2019, which are the days
prior to and after the solar eclipse. We also determined the upper and lower bounds for the baseline TEC based on the standard
deviation of IRI model output for 18–25 December 2019.

Figure 9 shows the TEC time series plots for 25–27 December 2019 at each of the 4 designated checkpoints. In these time
series plots, the observed TEC values (red curves) are compared to the normal baseline (cyan curves) that represent the diurnal
TEC variation in the absence of solar eclipse. Over all of these 4 checkpoints, the TEC responded to the solar eclipse in
relatively similar fashion, with a significant reduction during the eclipse. This TEC reduction was subsequently followed by
a recovery that went past the end of the eclipse. In general, the rate of TEC reduction was different for different checkpoints.
More notably, although the TEC reduction at all of these 4 checkpoints happened monotonically, the TEC recovery process
was not as monotonic. At certain checkpoints (KTB and GDU), the TEC recovery was quite uneven and occurred in graduated
steps. Further, the minimum TEC at each checkpoint was reached with different time delay relative to the moment of maximum
eclipse. The details of these features can be seen more clearly in the $\Delta$TEC time series.

Figure 10 shows the $\Delta$TEC time series plots for 25–27 December 2019 at each of the 4 designated checkpoints. In compari-
son to the TEC variation on non-eclipse days, the reduction in TEC during the solar eclipse at these 4 locations was quite large.
The depth of the TEC reduction valley was generally different for each checkpoint. Over the checkpoint KTB, the depth of the
TEC reduction valley was $-4.2$ TECU. Over the checkpoint GEC, the depth of the TEC reduction valley was $-4.5$ TECU.
Meanwhile, the depth of the TEC reduction valley over the checkpoints GDU and PTK was $-5.0$ TECU and $-5.5$ TECU,
respectively. Among the 4 designated checkpoints, the smallest reduction was found over KTB. In general, the TEC reduction
valley appears to be deeper for checkpoints located further east. In addition, we can also discern the non-monotonicity of the
TEC recovery process over these 4 designated checkpoints, which was exhibited with varying degree.

A full summary of the observed reduction in TEC over the 4 designated checkpoints as a response to the eclipse, with some
275 comparison to the corresponding reduction in foF2 (for the KTB and PTK checkpoints), is presented in Table 1. The tabulated
information includes the amount of reduction, the time delay between maximum eclipse and the lowest point in the reduction
valley, as well as the average reduction and recovery rates.

In order to gain a better understanding on the ionospheric response to the solar eclipse beyond the examination of TEC and
$\Delta$TEC time series over fixed checkpoints, we also conducted a more elaborate analysis using a set of cross-sectional cut lines.
The TEC, $\Delta$TEC, and TEC perturbation (TECP) data along these cross-sectional cut lines are assembled into keogram plots,
which enable us to reveal greater complexity in the pattern of ionospheric response to the solar eclipse. The results from this
analysis are presented below.

Figure 11 shows a geographical map depicting a set of cross-sectional cut lines that were used for the analysis. One is
oriented along the eclipse trajectory, which we refer to as the *parallel evaluation arc*. Meanwhile, the other four are oriented
perpendicular to the eclipse trajectory, which we refer to as *x-cut lines* #1 – #4. These cross-sectional cut lines were chosen
according to their proximity to the eclipse trajectory and the availability of GNSS receiver stations. Further, x-cut lines #2 and
#4 were chosen to include the coordinates of the Kototabang and Pontianak ionosondes.





Figure 12 shows the keogram plots for TEC, ΔTEC, and TECP as a function of UTC and longitude along the parallel evaluation arc. The TEC, ΔTEC, and TECP values are indicated using colormaps. Figure 12a depicts 2-day (2 × 24 hour) worth of data on 25–26 December 2019, whereas Figure 12b shows a magnification around the period of the solar eclipse. On the left panel of Figure 12a, the normal diurnal pattern of TEC variation at various longitudes in absence of solar eclipse can be recognized from the first half of the data (covering 25 December 2019). During the solar eclipse (period marked with the dashed/dotted lines on 26 December 2019), the pattern of TEC reduction was highly visible since at this time of day, TEC would normally have neared its daily maximum level. The left panel of Figure 12b offers a greater visual detail. The pattern of eclipse-related TEC reduction along the parallel evaluation arc was generally found to be uneven. In the longitude span from 90°E to 102°E during the eclipse, TEC was considerably lower than that in other longitudes. This uneven pattern in TEC reduction is likely due to the difference in solar local time at various points along the parallel evaluation arc when the solar eclipse happened.

On the center panel of Figure 12a, we show the ΔTEC as a function of UTC and longitude along the parallel evaluation arc. Similar to the case of the four fixed checkpoints, the ΔTEC values were obtained by subtracting a smooth baseline from the absolute TEC values. However, in this case the smooth baseline for each individual longitude was determined by taking the average of the corresponding TEC data from the entire month (excluding 18, 19, 20, and 26 December 2019 due to geomagnetic activity and the solar eclipse). During the initial phase of the eclipse, the decrease in ΔTEC did not immediately take place. Not until nearing the maximum eclipse did ΔTEC started to drop, which eventually reached approximately -6 TECU at its lowest. The negative ΔTEC remained until the end of the eclipse. The center panel of Figure 12b offers a greater visual detail. We also recognized that the pattern of reduction in ΔTEC was highly uneven. The duration of the ΔTEC reduction valley varied slightly across different longitudes. But more notably: over the longitude span from 96°E to 107°E, the ΔTEC reduction valley was visibly not as deep as in other locations. This "shallow ΔTEC valley" occurred over the Sumatra island, which is a major landmass traversed by the solar eclipse. The precise cause of this shallow valley is uncertain at this point.

On the right panel of Figure 12a, we show the TECP as a function of UTC and longitude along the parallel evaluation arc. The TECP was derived through a detrending process. But unlike ΔTEC previously, here the TEC data detrending was performed on the IPPs, before we spatially interpolated the detrended values onto the cross-sectional cut lines. The TECP values on individual IPPs were obtained by subtracting the 30-minute running average. After detrending, the TECP values were then spatially interpolated onto the parallel evaluation arc using the IDW technique. Generally speaking, there are wavelike fluctuations in the TECP data at nearly all times of day. However, there were recognizable changes in the TECP pattern that matched the period of the solar eclipse. These distinct wavelike fluctuation patterns persisted until the end of the eclipse. The characteristics of the wavelike fluctuation patterns varied during different phases of the solar eclipse. From the start of the eclipse until the maximum eclipse, there was a wavelike fluctuation pattern where the average TECP was biased toward +ve polarity. At the maximum eclipse, the TECP values dipped sharply to become negative. Finally, from the maximum eclipse until the end of the eclipse, there was another wavelike fluctuation pattern but this time with the average TECP slightly biased toward −ve polarity.





Figure 13 shows keogram plots for the TEC values as a function of UTC and latitude along the four x-cut lines, covering time interval 01:00–09:00 UTC on 25–27 December 2019 at different rows. Reduction in the TEC values can be seen along each of the four x-cut lines during the solar eclipse. At each x-cut line, however, there was different time delay from the start

of the eclipse (C1) until a significant decrease in TEC started to happen. This time delay ranged between 1 and 2 hours. The TEC reduction occurred predominantly to the north of the greatest eclipse point for each x-cut line.

Figure 14 shows keogram plots for the ΔTEC values as a function of UTC and latitude along the four x-cut lines, covering time interval 01:00–09:00 UTC on 25–27 December 2019 at different rows. The patterns of TEC reduction along each of the four x-cut lines during the eclipse are more clearly visible in this case. These keogram plots confirm that the TEC reduction

occurred more predominantly to the north of the greatest eclipse point for each x-cut line. This is signified by the ΔTEC in the northern part of the x-cut lines being more negative than that in the southern part.

Figure 15 shows keogram plots for the TECP values as a function of UTC and latitude along the four x-cut lines, covering time interval 03:00–08:00 UTC on 25–27 December 2019 at different rows. At the time of maximum eclipse, the TECP values in the keogram formed a prominent trough that is oriented parallel to the maximum eclipse line. Meanwhile along x-cut #3, a

335 convex bulge with positive TECP polarity was seen between latitudes $-5°$ and $0°$ just after the C1 phase. The convex bulge lasted for approximately 30 minutes. The tip of this bulge occurs started at latitude $-2°$ and the rest of this bulge then widened north/south, resembling a left-facing wavefront. Between C1 and C4, there were a number of striped patterns that indicate oscillations in the TECP data. The contours of these striped patterns were oriented roughly parallel to the contours of the C1 and C4 epoch lines. These wavelike fluctuations might be caused either by non-uniformity (wrinkles) embedded in the solar

EUV irradiance patterns, or by acoustic-gravity waves (AGWs) manifested as traveling ionospheric disturbances (TIDs) due to eclipse-related localized cooling in the Earth's atmosphere. The closer the alignment of the stripe patterns to the C1/max/C4 epoch lines, the more likely they are to be associated with the EUV modulation mechanism. The further away the striped patterns were from alignment with the C1/max/C4 epoch lines, the more likely they are to be associated with AGW/TID.

### 3.3 Solar EUV Illumination Variability

During the 26 December 2019 annular solar eclipse over the Indonesian sector, the Moon's umbra moved from west to east with a ground speed of approximately 1.1 km/s (Espenak, NASA GSFC, 2019). Despite this high speed, temporal and spatial variation in the solar EUV irradiance over eclipse-affected areas still happened gradually in stages. With a finite width of the eclipse central path (i.e. the umbra) of approximately 117-120 km, partial and annular eclipse also occurred consecutively along the eclipse central path in a gradual manner. Using the Laplacian as the main metric, we are able to capture the inhomogeneity

in the solar EUV irradiance. This characterization was performed by computing the Laplacian of the solar EUV irradiance at various point locations at each epoch, which enabled us to map its full spatio-temporal evolution.

Figure 16 shows the pattern of the Laplacian of solar EUV irradiance calculated over a region that spans from longitude 70°E to 150°E and from latitude 30°S to 30°N, evaluated at different times during the passage of the solar eclipse. From these snapshots, we can see the spatial variation of solar EUV irradiance that was present during the eclipse, which generally

resembles the outline of the lunar shadow as it traversed across the region. Moreover, there were also signatures of ARs on





the Sun's surface in the spatial inhomogeneity of solar EUV irradiance received on Earth. They appear as smaller ring patterns within the greater outline of the lunar shadow in the EUV Laplacian map. All of these patterns moved across the region with velocity matching that of the lunar shadow.

Figure 17 shows the calculated spatio-temporal pattern of the EUV Laplacian along the the parallel evaluation arc. From this
contour/colormap plot, we can see how the solar EUV irradiance variation was spatio-temporally consistent with the timing of various eclipse phases. Outside of the eclipse period, there was no fluctuation in the Laplacian because all the ARs would have been fully visible from every point on Earth that was facing the Sun.

Figure 18 shows contour/colormap plots of the Laplacian of the solar EUV irradiance variation along x-cut lines #1 to #4 as a function of time and latitude, which exhibit some of the same basic patterns. As in the case of the parallel evaluation arc, there
was no fluctuation in the Laplacian outside of the eclipse period. One major difference here is that there were a set of distinct features around the greatest eclipse point for each x-cut line, which appear as pairs of pod-shaped blobs that sandwiched the annularity region. In Figures 18c and 18d, they resembled a macaron around the annularity region.

Although the basic outlines of the Laplacian of solar EUV irradiance found in Figure 17 and 18 are generally consistent with the TECP profiles shown in Figures 12 and 15, there are some important differences in the details of the patterns. We found
that some of the large structures observed in the TECP profile were not part of the EUV Laplacian profile. Further, several fine structures that appeared in the EUV Laplacian profile did not actually materialize in the TECP profile. More detailed discussion regarding the relations between the TECP and EUV Laplacian profiles is presented in Section 4.

## 4   Discussion

Throughout December 2019, the most disturbed geomagnetic condition occurred on 18–20 December 2019, with lowest Dst
index of $-28$ nT (World Data Center at Kyoto University) and highest Kp index of $4^-$ (GFZ Potsdam). During the solar eclipse day (26 December 2019), the geomagnetic condition was quiet with lowest Dst index of $-6$ nT and highest Kp index of $2^+$. Thus, in general the effect of geomagnetic activity may be neglected in the analysis of the solar eclipse effects on the ionosphere. The F10.7 solar flux index on the eclipse day was approximately 70 sfu (NASA OMNIWeb Database), and this value was quite stable during $\pm 3$ days prior to and after the solar eclipse day. There was no major changes in the background
solar flux that may significantly confound the solar eclipse effect.

The ionospheric time delay between the moment of maximum eclipse and minimum foF2 over KTB was 34 minutes, and that over PTK was 36 minutes. This time delay is shorter than the time delay from a number of other studies: Jose et al. (2020) reported a time delay of 1 hour, Adeniyi et al. (2007) reported a time delay of 40 minutes, Dear et al. (2020) reported a time delay of 1 hour, and Bravo et al. (2020) reported time delays of 4, 40, and 62 minutes. The shorter time delay may be caused
by the longer eclipse duration as a whole. The overall eclipse duration over KTB was 4 hours, and that over PTK was 3 hours 47 minutes. Meanwhile, the duration of solar eclipse events studied by Adeyini et al. (2007) and Dear et al. (2020) was about 2 hours. This difference in the overall eclipse duration may be the reason behind the significant difference in time delay.





The average rate of NmF2 (foF2) reduction over KTB and PTK was 43.0 el cm$^{-3}$/s (1.03 MHz/hour) and 20.64 el cm$^{-3}$/s (0.51 MHz/hour), respectively. The relative foF2 reduction with respect to baseline level was 23.2% over KTB (where maximum obscuration was 91%) and 22.4% over PTK (where maximum obscuration was 93%). This relative reduction is slightly smaller in magnitude than the eclipse-induced reduction reported in the past by Adeniyi et al. (2007) of 54%, and that by Dear et al. (2020) of 30%–40%. This difference may have been influenced by the type of solar eclipse that occurred and the actual local time of the solar eclipse event. The 26 December 2019 annular solar eclipse occured later in the day (10:20–14:30 LT) when compared to the eclipse event investigated by Adeniyi et al. (2007) which occurred between 09:00–11:00 LT, and that investigated by Dear et al. (2020) which occurred between 08:46–11:40 LT. The local time of the eclipse event mattered since the closer we are to noontime, the stronger the equatorial fountain effect tends to be. This proclivity is also mimicked by the strength of equatorial electrojet (EEJ) at noon, which is stronger than the pre-noon EEJ. As such, a stronger plasma flow associated with the equatorial fountain effect at midday might affect the amount of eclipse-induced foF2 reduction by its influence on the transport process. The 26 December 2019 annular solar eclipse occurred around noontime, during which the equatorial fountain may have lessened the amount of foF2 reduction to some degree since both KTB and PTK are located near the southern crest of the equatorial ionization anomaly (EIA).

The average recovery rate of NmF2 value over KTB and PTK was 23.04 el cm$^{-3}$/s and 19.9 el cm$^{-3}$/s, which occurred during time interval 05:45–08:45 UTC and 06:20–08:55 UTC, respectively. The respective time delay between the end of eclipse and the end of recovery phase over KTB and PTK was 97 minutes and 83 minutes. The relatively long time delay until the completion of the recovery phase might be caused by the additional influence of transport and diffusion process that slowed down the rate of plasma accumulation from direct photoionization.

The observed foF2 (and NmF2) reduction and recovery over PTK in general exhibited a symmetrical pattern. Meanwhile over KTB, the pattern was not so symmetrical: the rate of foF2 reduction was steeper than the rate of foF2 recovery. Based on the findings reported in past studies (Farges et al., 2001; Adeniyi et al., 2007; Goncharenko et al., 2018), symmetrical foF2 reduction/recovery pattern is more common. Hence, the pattern observed over KTB during this solar eclipse event was somewhat abnormal. Considering that the foF2 recovery rate over KTB was quite similar to that over PTK, the abnormality over KTB can be attributed to the foF2 reduction rate. In other words, the foF2 reduction rate over KTB was steeper than normally expected. The reason for a steeper foF2 reduction rate over KTB was most likely the higher initial foF2 (and NmF2) value at the start of the eclipse, which resulted in a higher recombination rate in the absence of photoionization, viz. $dN/dt = -\alpha N^2$ in the ionization balance equation (Rishbeth, 1963, 1968; Rishbeth and Garriott, 1969; Adeniyi et al., 2009).

In terms of changes in foF1 during the eclipse, the time delay between maximum eclipse and minimum foF1 over KTB and PTK was 9 minutes and 16 minutes, respectively. This relatively short response time confirms that the ionospheric F1 layer is dominated by production/loss mechanism involving direct photoionization and recombination process, and it is not affected significantly by transport process (Farges et al., 2001; Jose et al., 2020).

The relative foF1 reduction with respect to baseline level was 33% over KTB, and 21.5% over PTK. This relative foF1 reduction is similar to that reported by Jose et al. (2020), which showed a relative foF1 reduction of 29% over Trivandrum, India (8.5°N 77°E; 0.5°N MLAT) during the 15 January 2010 annular solar eclipse when the solar cycle was heading away





from a minimum towards a maximum. In contrast, Adeniyi et al. (2007) showed a relative foF1 reduction of 68% over Ilorin, Nigeria (8.53°N 4.57°E; 4.1°S MLAT) during the 29 March 2006 total solar eclipse during a minimum phase of the solar cycle.

The significant difference with Adeniyi et al.'s (2007) findings might be due to the different type of solar eclipse that occurred. The average rate of foF1 reduction over KTB was $-16$ el cm$^{-3}$/s, and that over PTK was $-29$ el cm$^{-3}$/s. The much steeper foF1 reduction rate over PTK was most likely due to a higher initial foF1 value, which resulted in a higher recombination rate ($dN/dt = -\alpha N^2$) in the absence of photoionization (Cheng et al., 1992; Adeniyi et al., 2009).

The average rate of foF1 recovery over KTB and PTK was of 44.0 el cm$^{-3}$/s and 15.49 el cm$^{-3}$/s which occurred during

time interval 05:20–06:25 UTC and 06:00–07:20 UTC, respectively. The difference between the rates of foF1 recovery over these two locations may look striking, with nearly 3:1 ratio. However, considering the spread of data points (2.90–3.53 MHz) near the minimum foF1 value over KTB stations, the registered recovery rate of 44.0 el cm$^{-3}$/s is actually associated with a fairly large range of uncertainty (plausible range of recovery rate: 31–44 el cm$^{-3}$/s). Thus, the difference in the average foF1 recovery rate between KTB and PTK may in fact be milder with a ratio closer to 2:1, similar to the difference in foF1 reduction

rate between PTK and KTB.

The foF2 time series on 26 December 2019 showed an overshoot during the recovery phase beyond the baseline level in the time interval 08:45–10:30 UTC (15:45–17:30 LT) over KTB, and in the time interval 09:05–10:30 UTC (16:05–17:30 LT) over PTK. The foF2 overshoot had a magnitude of 0.5–0.7 MHz above the baseline level. This overshoot might have been caused by an inward shift of the EIA crest position during the post-eclipse period, after an outward shift that happened earlier during

the eclipse (Aa et al., 2020). The shift of the EIA crest back to its original position happened simultaneously with the local recovery of foF2 by the restored photoionization, causing foF2 to overshoot the baseline value. A similar but less prominent feature was also observed in terms of the foF1 parameter, possibly due to the same reason. Further, we note that an overshoot feature in foF2 values was also observed during the 21 August 2017 solar eclipse over a midlatitude region (Goncharenko et al., 2018).

Examination of foF1 time series on 26 December 2019 over PTK revealed a momentary increase in foF1 value before it eventually decreased in response to the solar eclipse. The magnitude of the aforementioned increase was only ∼0.25 MHz, with a duration of ∼30 minutes. This particular feature is similar to past findings reported by Anastassiades and Moraitis (1968) and Jonah et al. (2020). This peculiar feature is thought to be caused by diffusion of electrons from the topside altitudes above hmF1 as the electron temperature remains greater than the neutral temperature during a solar eclipse. It is believed that this

feature can only occur for solar eclipse event that happens in the minimum phase of the solar cycle. The said feature was not found in the foF1 time series data over KTB.

In terms of general timing, we note here that the eclipse-related foF1 reduction and recovery occurred completely within the eclipse phase; whereas the eclipse-related foF2 reduction and recovery extended well beyond the eclipse period. This pattern reaffirms the established concept that there are different plasma production and loss mechanisms operating in the ionospheric

F1 and F2 layers (Rishbeth and Garriott, 1964, 1969; Rishbeth, 1968; Hargreaves, 1992). Plasma density in the ionospheric F1 layer is largely determined by photoionization and recombination only. Meanwhile, plasma density in the ionospheric F2 layer is influenced not only by photoionization and recombination, but also by transport process.





Over KTB, hpF2 was observed to decrease roughly 2 minutes after the start of the eclipse. Meanwhile over PTK, hpF2 started to decrease roughly 10 minutes after the start of the eclipse. The "downward motion" of hpF2 occurred for approximately 95
minutes over KTB (a 50 km overall descent), and 120 minutes over PTK (a 136 km overall descent). The observed decrease in hpF2 is probably only a "downward spatial relocation" of the altitude where the greatest electron density was found, as the general shape of the electron density profile changed during the eclipse. Without measurements of Doppler shift, it is hard to ascertain whether an actual downward plasma motion was present.

Over KTB, hpF2 started to rise 19 minutes after the maximum eclipse; and hpF2 reached its highest value of 682 km (220
465   km higher than the baseline height) at approximately 06:40 UTC (13:40 LT). Later on, hpF2 dropped back to the baseline height at approximately 07:00 UTC (14:00 LT) while still within the eclipse period. Meanwhile over PTK, hpF2 started to rise 11 minutes after the maximum eclipse; and hpF2 reached its highest value of 747 km (326 km higher than the baseline height) at approximately 07:35 UTC (14:35 LT). Later on, hpF2 value returned to the baseline height at approximately 08:10 UTC (15:10 LT). Possible cause of this increase in hpF2 includes (1) changing shape of electron density profile during the recovery
phase that resulted in a "spatial relocation" of the altitude with largest electron density, and (2) an actual upward movement of the ionospheric F2 layer due to a polarization electric field that formed between west and east sides of the eclipse-affected region, which induced a vertical $\mathbf{E} \times \mathbf{B}$ drift (Jakowski et al., 2008; Le et al., 2009; Adekoya et al., 2015).

With regard to the TEC observation data, there are a few things of note. These include the spatial inhomogeneity in the depths of the TEC reduction valley along the main eclipse trajectory, the spatial inhomogeneity in the depths of the TEC reduction
valley in the direction perpendicular to the main eclipse trajectory, and the general relations between the TEC variation during the solar eclipse and the detailed shape of the solar EUV illumination profile.

Based on the observed ΔTEC along the parallel evaluation arc (cf. second column of Figure 12), we can discern that the ΔTEC values between longitude 96°E to 106°E were not as deep as ΔTEC at other locations that were also along the eclipse trajectory, even though they all roughly experienced the same level of solar obscuration and eclipse duration. This feature can
only be seen clearly in the ΔTEC data, but not in the absolute TEC data as the absolute TEC contains a mixture of both the diurnal variations and the eclipse response. The definitive reason behind this "shallow ΔTEC valley" is not so clear, but we believe that it involved a physical mechanism, and it was not simply an instrumental effect. The GNSS receiver stations in the region are not uniformly distributed, but the spatial distribution of receiver stations are not expected to affect TEC bias calibration.

In addition to the variation in the depth of ΔTEC valley in east-west direction along the eclipse trajectory, there was also a notable north-south variation in the TEC and ΔTEC valley depths along the four x-cut lines as a function of orthogonal distance from the main eclipse trajectory (cf. Figures 13 and 14). Somewhat counterintuitively, the lowest TEC and ΔTEC values actually occurred to the north of the eclipse trajectory, even though the lunar shadow was coming from a southerly direction with solar zenith angle of $\sim 23°$ during the eclipse event. Hence this effect cannot be explained simply by the slight
obliquity of the lunar shadow relative to vertical direction. Instead, this effect was probably induced by the equatorial fountain effect which transports plasma from magnetic equator area (located north of the eclipse trajectory) to regions at $\pm 15°$ MLAT, effectively providing buffer for the TEC in the area south of the eclipse trajectory. As a result of this siphoning, the area north





of the eclipse trajectory was left with a greater deficit in TEC. Such a role of the fountain effect during a solar eclipse over low-latitude region is consistent with findings reported by Aa et al. (2020).

Further, there are some temporal inhomogeneity and fluctuation patterns in the TEC observation data as well. The TEC and $\Delta$TEC data at various observation points indicate that the rate of TEC decrease and increase during the eclipse was generally not symmetrical. Overall, the rate of TEC decrease is greater in magnitude than the rate of TEC increase/recovery. This asymmetric pattern may bear some relations to the shape of the solar EUV illumination profile, which itself also shows some asymmetry. In addition, it is also conceivable that kinks/wrinkles in the solar EUV illumination profile could manifest

themselves in the TECP values. The effects of inhomogeneity and kinks/wrinkles in the solar EUV illumination profile during solar eclipse had been previously discussed in Mrak et al. (2018). Nevertheless, inhomogeneity in the solar EUV illumination profile may not always fully manifest itself in the TEC patterns. Here we discuss several aspects of the TEC data that are pertinent to the potential role(s) of inhomogeneous solar EUV illumination profile during this eclipse event.

    Predominately, we direct our attention to a momentary increase (of ~10 minute duration) in the solar EUV illumination

around the maximum phase of the solar eclipse (cf. Figure 4b). The appearance of this brief yet distinct peak in the solar EUV illumination profile near the maximum eclipse phase probably arose due to the fact that it was an annular solar eclipse. In an annular solar eclipse, significant solar illumination would be present during the maximum eclipse phase with the rim of the solar disk left unobscured. Such an illumination by the rim of the solar disk would not be as prominent for a total solar eclipse. Despite the distinguishing form of this mini-climax in EUV illumination, there was no corresponding bump in the TEC or

$\Delta$TEC values in response to it. The TEC or $\Delta$TEC data had been gridded with 3-minute time resolution, which should provide sufficient sampling to capture such a bump in TEC or $\Delta$TEC had it actually existed. Most likely, the duration of exposure was too short for the TEC to respond in ways that would closely mimic the aforementioned mini-climax.

    The next set of aspects to be discussed are associated with the TECP values. Fluctuation patterns in TECP values that correlate with the eclipse events along the parallel evaluation arc and the four x-cut lines can be seen in Figures 12 and 15.

Fluctuations in TECP values can be associated with TIDs that are driven by AGWs in the neutral part of the upper atmosphere. However, it is also possible that fluctuation patterns in TECP values were due to modulation by kinks/wrinkles in the solar EUV illumination profile. Fluctuation patterns in TECP due to these two factors will have different characteristic properties. If the TECP fluctuation originated from solar EUV modulation, the resultant TECP fluctuation patterns should have a high degree of similarity with patterns that are visible in the calcuated Laplacian EUV profile (cf. Figures 17 and 18).

The calculated Laplacian EUV profile in Figure 17 showed prominent feature around the maximum eclipse phase. This feature was oriented parallel to the C1/max/C4 epoch lines that ran from west to east. When compared with the TECP data (cf. third column of Figure 12), there were similar features that also appeared along the maximum eclipse epoch line. In this aspect, there is a notable similarity between the Laplacian EUV pattern and the observed TECP pattern. Likewise, the TECP data in Figure 15 show a number of stripes that are well-aligned with their respective C1/max/C4 epoch lines, matching the general

orientation of the surface texture of the Laplacian EUV pattern shown in Figure 18. These characteristics are consistent with fluctuations due to direct modulation by inhomogeneous solar EUV illumination.



However, there were other features that differ significantly between the Laplacian EUV pattern and the TECP observations. The surface texture of the Laplacian EUV in Figure 17 ran generally parallel to the C1/max/C4 epoch lines. Meanwhile in the TECP data (cf. third column of Figure 12), there are fluctuation patterns between longitudes 92°E and 102°E around 06:00 UTC with wavefronts that are oriented sideways, deviating considerably from the orientation of the C1/max/C4 epoch lines. In addition, the large bulge in TECP that appeared between C1 and maximum eclipse along x-cut #3 (cf. Figure 15b) was not predicted by the Laplacian EUV calculation (cf. Figure 18c). This TECP feature is more consistent with AGW/TID. Further, there was a special feature in the Laplacian EUV pattern (cf. Figure 18 all panels) where pairs of pod-shaped blobs appeared around the moment of maximum eclipse, sandwiching the annularity region. This particular pattern was present in the calculated Laplacian EUV profile for all x-cut lines, and hence not an isolated incidence. However, no associated feature was found in the TECP data. The absence of the sandwich feature in the TECP data reveals some limitations in how much the inhomogeneous solar EUV illumination can directly modulate the ionosphere. Our findings on emergent features of wavelike perturbations in TEC data during the 26 December 2019 eclipse are consistent with Barad et al. (2022), who found similar features from observations over the Indian region, while ours came from the Southeast Asian region and specifically over the Indonesian sector.

## 5   Summary and Conclusion

We have examined a number of ionospheric effects associated with the passage of an annular solar eclipse on 26 December 2019 over the Indonesian region. Observation data from ionosonde and ground-based GPS receiver instruments, accompanied with solar imagery data from the SDO AIA instrument, were analyzed in the study. The conclusions are as follows:

1. Ionosonde observations indicate reduction and subsequent recovery of ionospheric density during the solar eclipse, with the rates of reduction/recovery not necessarily symmetrical. The relative reductions of foF2 and foF1 were in the range of 22.4–23.2% and 21.5–33%, respectively. The rates of electron density reduction and recovery in the F2 layer were in the range of 20.6–43.0 el cm$^{-3}$/s (0.51–1.03 MHz/hr in terms foF2) and 19.9–23.0 el cm$^{-3}$/s (0.48–0.53 MHz/hr in terms of foF2), respectively.

2. Ionosonde observations indicate a delay between maximum eclipse and minimum foF2 (considerable time delay between 34–36 minutes), as well as between maximum eclipse and minimum foF1 (much shorter time delay between 9–16 minutes).

3. Ionosonde observations indicate that hpF2 descended (by 50–136 km) at the start of the eclipse. During the latter half of the eclipse, hpF2 was seen rising to reach 682–747 km (220–326 km higher than normal baseline). The apparent hpF2 rise started to happen ~11 minutes after the maximum eclipse, and ended ~90 minutes after the maximum eclipse.

4. Relative TEC reduction around the eclipse trajectory was in the 24.9–28.5% range. We found an interesting feature where the greatest TEC reduction occurred to the north of the eclipse trajectory, even though the lunar shadow came at



an angle from the south. It was probably facilitated by the equatorial fountain effect, which helped siphon plasma from geomagnetic equator (north of the eclipse path) and transport it toward the $\pm 15°$ MLAT area.

5. During the eclipse, wavelike fluctuations in the TECP data were found. Mechanisms associated with AGW/TID as well as due to direct modulation by non-uniform solar EUV illumination, each with its own characteristics, were considered. Some features in the TECP are characteristically more consistent with AGW/TID, while other features could be compatible with the solar EUV modulation effect based on analysis of SDO AIA solar images. However, a set of distinct solar EUV modulation patterns that had been predicted in the solar image analysis failed to manifest in TECP obser-

vations. It may indicate that the ability of non-uniform solar EUV illumination to modulate ionospheric plasma density configuration is highly sensitive to the particulars of the solar eclipse event.

*Data availability.*   Scaled ionogram parameters and processed GPS TEC data used in this study are made available via **https://doi.org/10.7910/DVN/ZUX(** Supplementary figures and tables are also provided in **https://doi.org/10.7910/DVN/ZUXCCK**.

*Video supplement.*   Animated sequence of solar AIA images with synthetic lunar disk masking, animated sequence of data maps showing

spatio-temporal evolution GPS TEC values on individual IPPs at 350 km altitude, and animated sequence of data maps showing spatio-temporal evolution of the Laplacian of solar EUV irradiance over the studied region can be accessed via **https://doi.org/10.7910/DVN/ZUXCCK**

*Author contributions.*   **Conceptualization:** JH, VD, AH, JM, AF, RP. **Ionosonde data acquisition:** AB, EE. **Ionosonde data processing & analysis:** JH,VD, AF, RP. **GPS data curation:** AH, AS. **GPS data processing & analysis:** RP, AH. **Solar EUV image data curation & analysis software development:** JM. **Methodology formulation:** RP, JH. **Supervision:** RP. **Manuscript writing - original draft:** JH, VD,

AH, JM, AF, RP. **Manuscript editing:** RP, VD.

*Competing interests.*   The authors declare that they have no conflict of interest.

*Acknowledgements.*   The authors thank the technical staff and ionosonde operators at the BPAA LAPAN Pontianak and Agam (Kototabang), as well as the BIG's INACORS network, for maintaining and operating the research equipment. The authors also thank Ecep Edi Hidayat for his help in manual scaling of raw ionogram data. The Dst index data can be accessed at **https://wdc.kugi.kyoto-u.ac.jp/dstdir/**. The Kp index

data can be accessed at **https://www.gfz-potsdam.de/en/kp-index/**. The F10.7 index data can be accessed at **https://omniweb.gsfc.nasa.gov/form/dx1.ht** The AIA SDO image data can be accessed at **https://sdo.gsfc.nasa.gov/data/aiahmi/**. Information on the BIG's INACORS network can be found at **http://inacors.big.go.id/sbc/**. Rezy Pradipta's time was partially supported by AFOSR grant FA9550-20-1-0313.



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



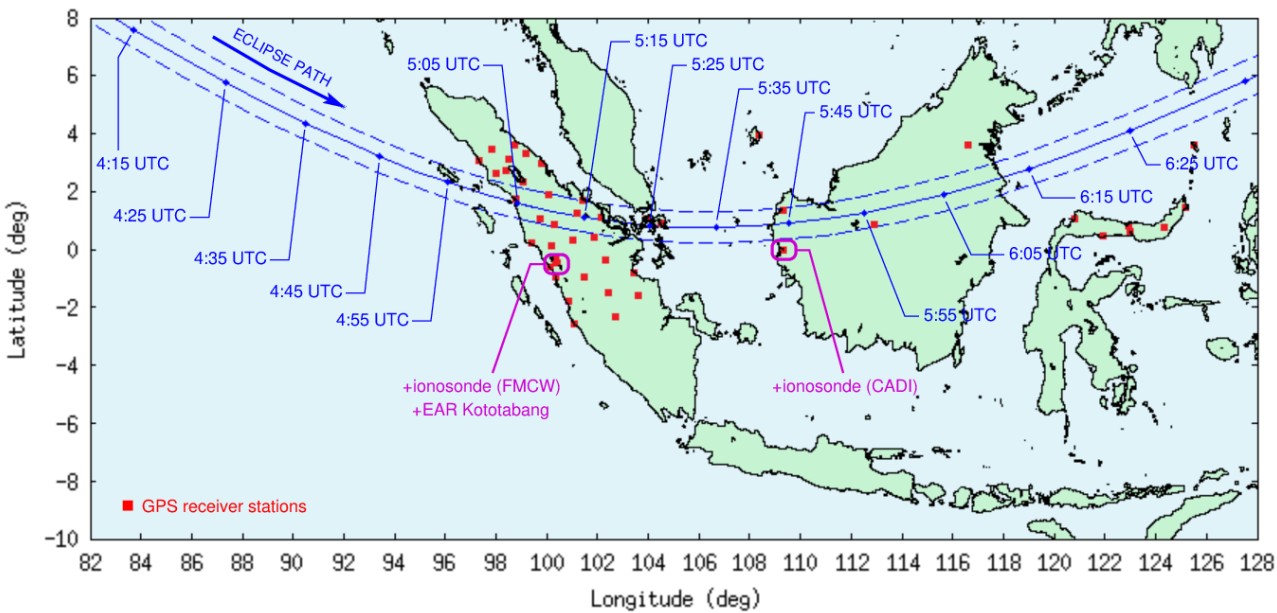

**Figure 1.** Geographical map showing the path of the 26 December 2019 annular solar eclipse over Southeast Asia, as well as a suite of ground-based radio diagnostic instruments used in the study.



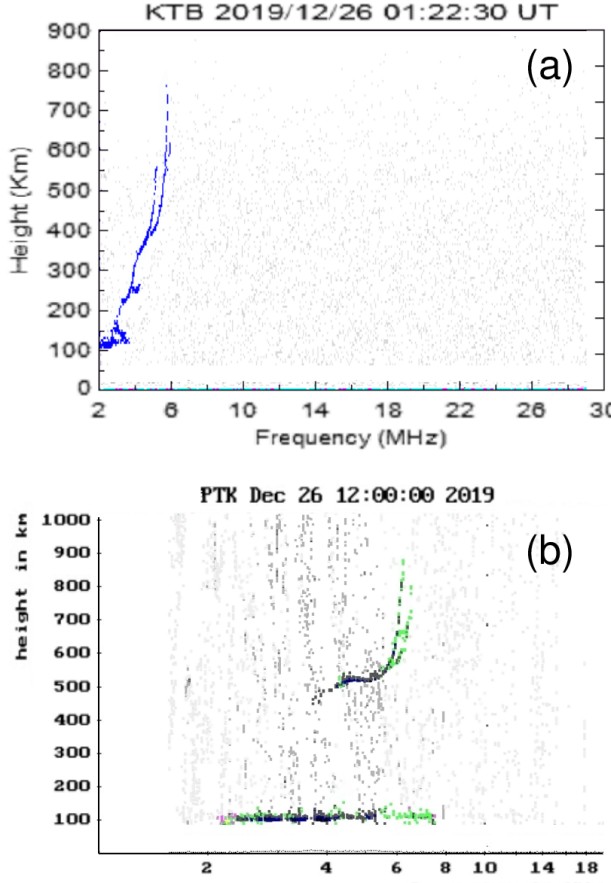

**Figure 2.** Sample ionograms from (a) Kototabang station at 01:22 UTC on 26 December 2019, and (b) Pontianak station at 12:00 LT (= 05:00 UTC) on 26 December 2019. Main return traces in these sample ionograms have been enhanced to make them stand out against high levels of background noise. Ionograms from these two stations were subject to manual scaling process in order to extract essential ionospheric parameters.



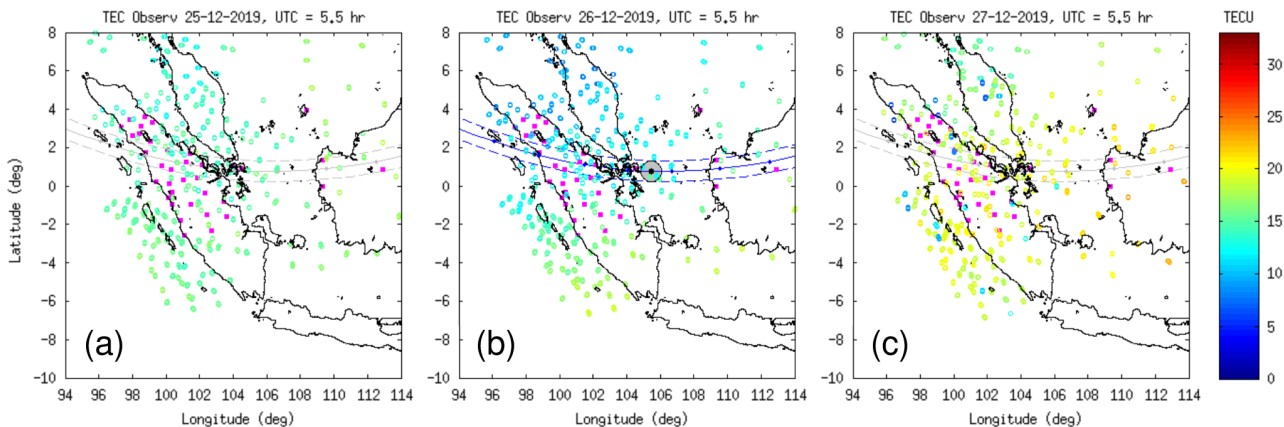

**Figure 3.** Geographical maps showing spatial distribution of the GPS IPPs at 350 km altitude at 05:30 UTC on (a) 25 December 2019, (b) 26 December 2019, and (c) 27 December 2019. The path of the annular solar eclipse on 26 December 2019 is shown as blue curve, and the location of eclipse annularity at this epoch is shown as a gray filled circle. Locations of ground-based GPS receiver stations are shown as magenta squares, and the TEC observation values are indicated on the individual IPPs using a colormap.



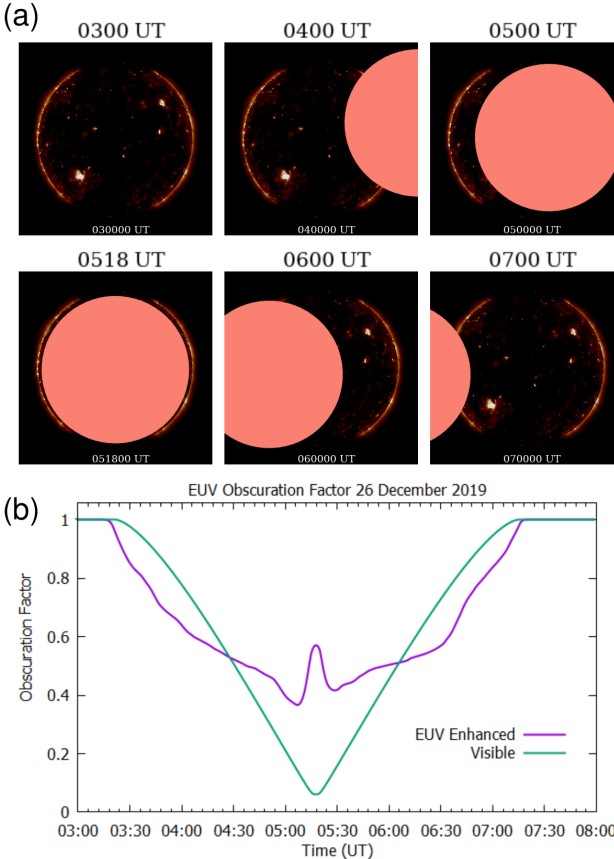

**Figure 4.** (a) Sample sequence of digitally masked 193 Å (19.3 nm) solar AIA images on 26 December 2019 to estimate the level of solar obscuration at a given point on the Earth's surface during the course of the solar eclipse. (b) Time series variation of the obscuration of solar irradiance at various wavelengths during the 26 December 2019 annular solar eclipse over Siak, Riau Province (1.01°N 102.25°E), calculated using this computational masking method.





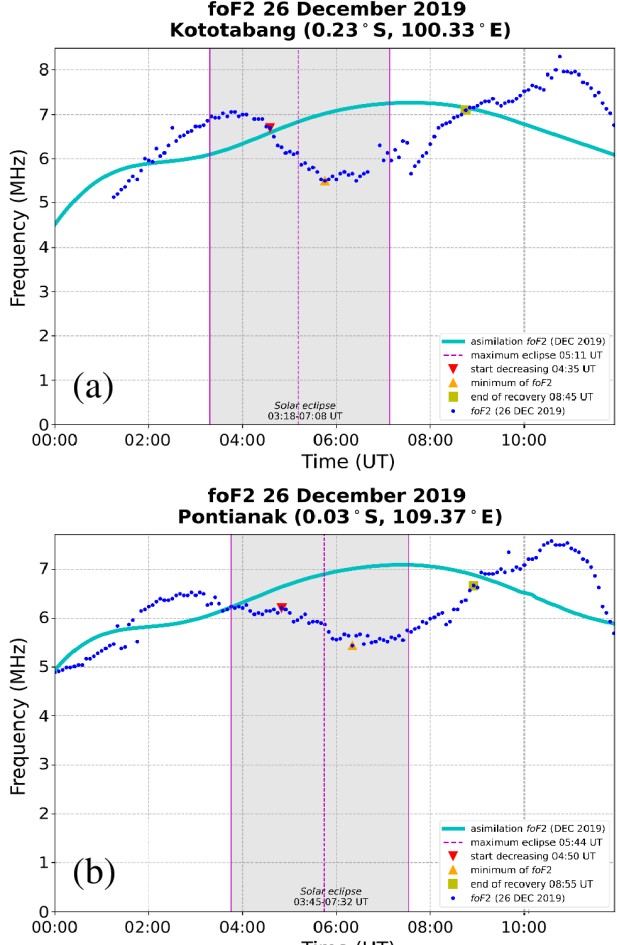

**Figure 5.** Time series plots of ionospheric F2 layer critical frequencies (foF2) from scaling of ionosonde measurements at (a) Kototabang and (b) Pontianak on 26 December 2019. Observation data are shown in blue, baseline foF2 level based on the December 2019 median is shown as cyan curves, and the solar eclipse period is indicated with gray bands.

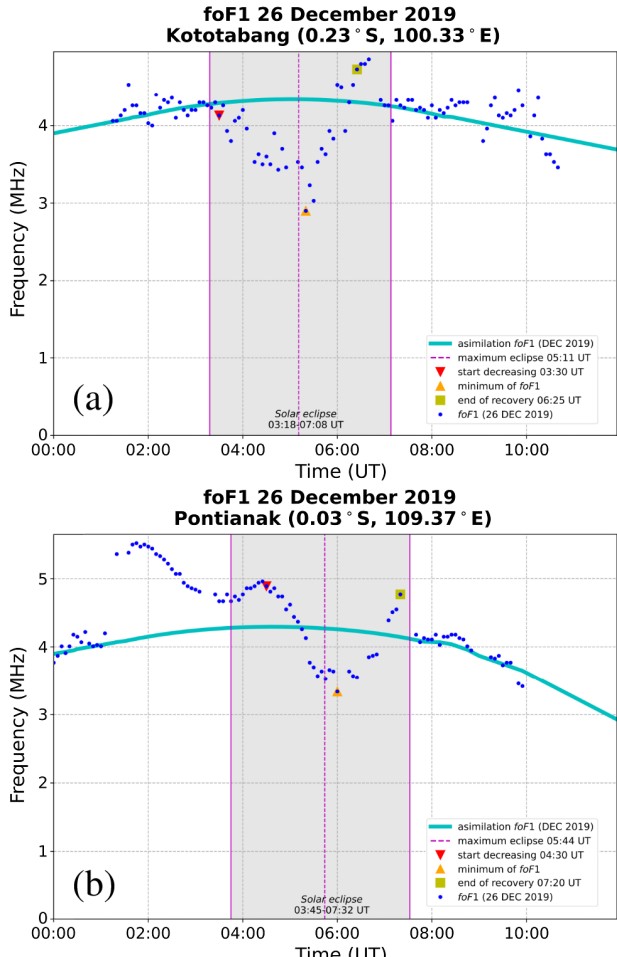

**Figure 6.** Time series plots of ionospheric F1 layer critical frequencies (foF1) from scaling of ionosonde measurements at (a) Kototabang and (b) Pontianak on 26 December 2019. Observation data are shown in blue, baseline foF1 level based on the December 2019 median is shown as cyan curves, and the solar eclipse period is indicated with gray bands.

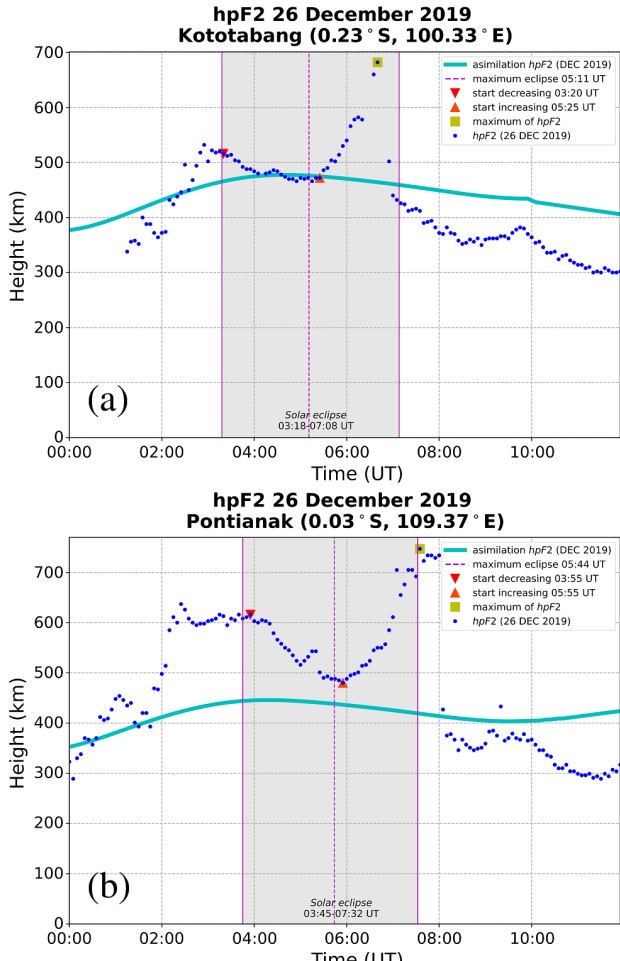

**Figure 7.** Time series plots of ionospheric height with maximum electron density (hpF2) from scaling of ionosonde measurements at (a) Kototabang and (b) Pontianak on 26 December 2019. Observation data are shown in blue, baseline hpF2 level based on the December 2019 median is shown as cyan curves, and the solar eclipse period is indicated with gray bands.



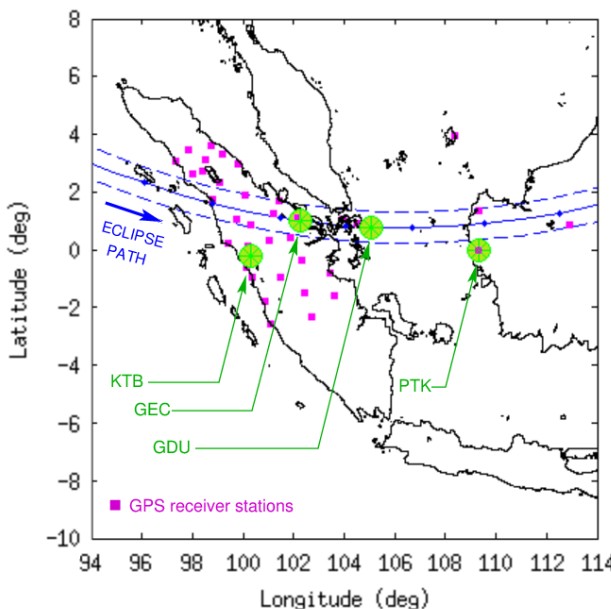

**Figure 8.** Geographical map showing a set of discrete check points for the TEC and $\Delta$TEC time series analysis shown in Figures 9 and 10. These fixed check points are over Kototabang (KTB; 0.20°S 100.32°E), point of greatest eclipse (GEC; 1.01°N 102.25°E), point of greatest annularity duration (GDU; 0.78°N 105.08°E), and Pontianak (PTK; 0.04°S 109.35°E). The path of the annular solar eclipse is shown on the map as blue curve, and locations of ground-based GPS receiver stations as magenta squares.





**Figure 9.** (a-d) Time series plots of GPS TEC at the 4 designated check points (KTB, GEC, GDU, and PTK) on 25-27 December 2019. Smooth background TEC levels based on the 25 and 27 December 2019 data, combined with IRI model runs on 18-25 December 2019, are shown in cyan. Observation data are shown in red. The solar eclipse period is indicated with gray bands.







**Figure 10.** (a-d) Time series plots of ΔTEC at the 4 designated check points (KTB, GEC, GDU, and PTK) on 25-27 December 2019, obtained by subtracting the smooth background TEC level from the GPS TEC observation data. The solar eclipse period is indicated with gray bands.





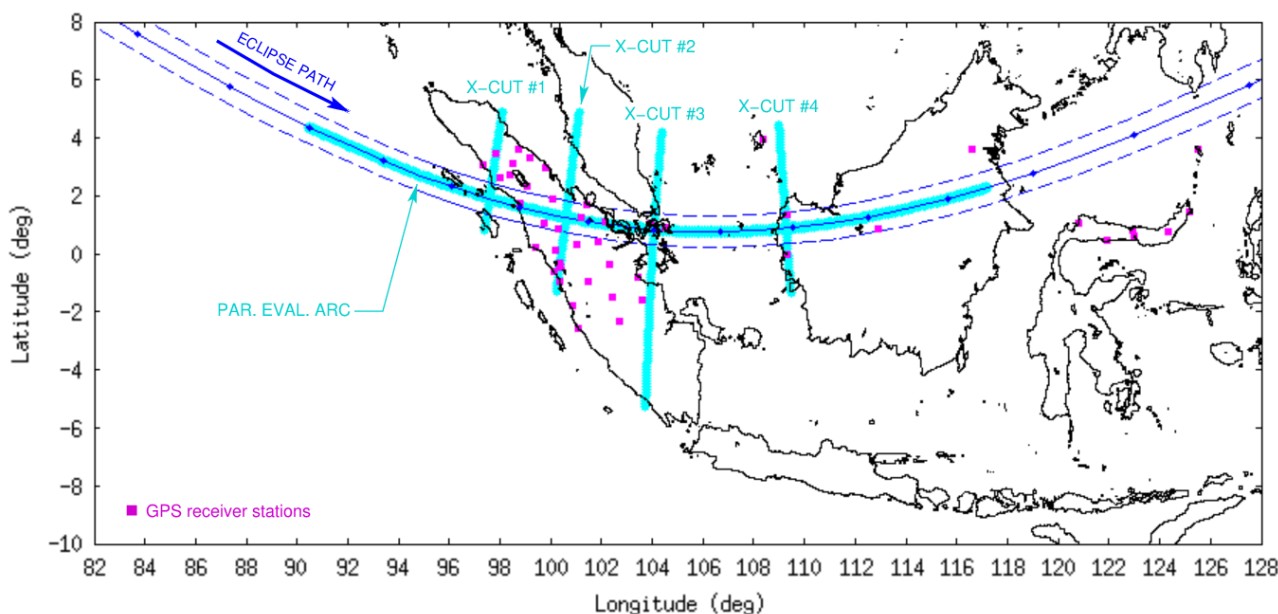

**Figure 11.** Geographical map showing a set of cut lines, along and perpendicular to the solar eclipse path, for an extended analysis of TEC variation in response to the eclipse. The cut line along the eclipse path is referred to as the *parallel evaluation arc*, and those perpendicular to the eclipse path are referred to as *x-cuts* #1 to #4.





**Figure 12.** Colormap plots of GPS TEC, ΔTEC, and TECP values as a function of time and longitude along the parallel evaluation arc (a) within a 48-hour time interval on 25-26 December 2019, and (b) within a magnified time interval around the solar eclipse on 26 December 2019. The start and end times of the eclipse (C1 and C4) as well as the moment of maximum eclipse are indicated by dashed and dotted curves on the plots.







**Figure 13.** Colormap plots of GPS TEC values as a function of time and latitude along x-cuts #1 to #4 on (a) 25 December 2019, (b) 26 December 2019, and (c) 27 December 2019. Time intervals are magnified around the solar eclipse on 26 December 2019. The start and end times of the eclipse (C1 and C4) as well as the moment of maximum eclipse are indicated by dashed and dotted curves on the plots. Circles indicate the moment and location of the greatest solar obscuration along each x-cut.





**Figure 14.** Colormap plots of ΔTEC values as a function of time and latitude along x-cuts #1 to #4 on (a) 25 December 2019, (b) 26 December 2019, and (c) 27 December 2019. Time intervals are magnified around the solar eclipse on 26 December 2019. The start and end times of the eclipse (C1 and C4) as well as the moment of maximum eclipse are indicated by dashed and dotted curves on the plots. Circles indicate the moment and location of the greatest solar obscuration along each x-cut.







**Figure 15.** Colormap plots of TECP values as a function of time and latitude along x-cuts #1 to #4 on (a) 25 December 2019, (b) 26 December 2019, and (c) 27 December 2019. Time intervals are magnified around the solar eclipse on 26 December 2019. The start and end times of the eclipse (C1 and C4) as well as the moment of maximum eclipse are indicated by dashed curves on the plots. Small bulge on the maximum eclipse line indicates the annularity region.





**Figure 16.** Sequential snapshots of spatial inhomogeneities in the solar EUV illumination pattern over Southeast Asia on 26 December 2019 as the annular solar eclipse traversed through the region. Contours and colormap indicate the absolute magnitude of the Laplacian of the solar EUV irradiance distribution over geographic latitude/longitude grid at 100 km altitude.





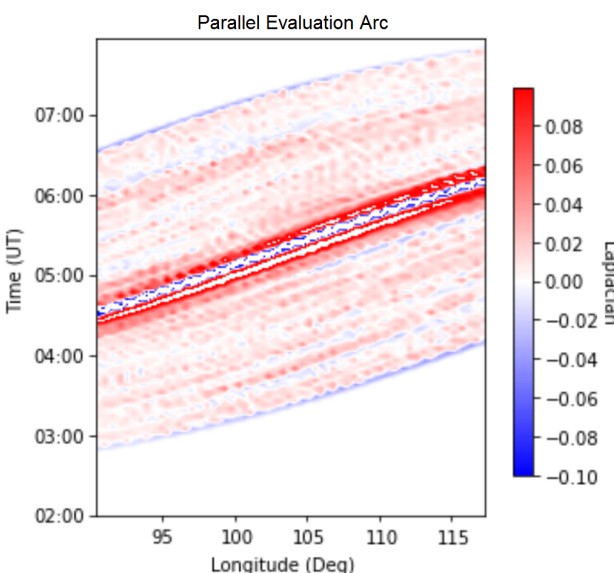

**Figure 17.** Contour/colormap plot of the Laplacian of the solar EUV irradiance level at 100 km altitude as a function of time and longitude along the parallel evaluation arc, within a magnified time interval around the solar eclipse on 26 December 2019.





**Figure 18.** (a-d) Contour/colormap plot of the Laplacian of the solar EUV irradiance level at 100 km altitude as a function of time and latitude along x-cuts #1 to #4, within a magnified time interval around the solar eclipse on 26 December 2019. Dashed circles highlight the characteristic pairs of pod-shaped blob patterns that appear in the vicinity of the annularity region.



**Table 1.** Summary of observed changes in foF2 and TEC at the 4 designated check points (cf. Figure 8) due to the passage of annular solar eclipse over Indonesia on 26 December 2019.

| Site Information | foF2 Reduction | Minimum foF2 | foF2 Recovery | TEC Reduction & Recovery |
|---|---|---|---|---|
| KTB<br>eclipse timing:<br>start 03:18 UTC<br>max 05:11 UTC<br>(obsc = 91.0%)<br>end 07:08 UTC | 04:35–05:45 UTC<br>(duration: 70 minutes)<br>initial foF2 = 6.70 MHz<br>$(5.5 \times 10^5$ el cm$^{-3}$)<br>foF2 reduction rate<br>= −1.03 MHz/hr<br>(−43 el cm$^{-3}$/sec) | min foF2 = 5.50 MHz<br>reached at 05:45 UTC<br>$\Delta$foF2 = −1.67 MHz*<br>(23.2% relative reduction*)<br>time lag = 34 minutes** | 05:45–08:45 UTC<br>(duration: 180 minutes)<br>final foF2 = 7.10 MHz<br>$(6.2 \times 10^5$ el cm$^{-3}$)<br>foF2 recovery rate<br>= +0.53 MHz/hr<br>(+23.0 el cm$^{-3}$/sec) | min $\Delta$TEC = −4.34 TECU*<br>reached at 05:51 UTC<br>(24.9% relative reduction*)<br>time lag = 40 minutes**<br>reduction rate = −4.06 TECU/hr<br>recovery rate = +3.13 TECU/hr |
| GEC<br>eclipse timing:<br>start 03:23 UTC<br>max 05:18 UTC<br>(obsc = 94.1%)<br>end 07:14 UTC | —— N/A —— | —— N/A —— | —— N/A —— | min $\Delta$TEC = −4.58 TECU*<br>reached at 05:42 UTC<br>(27.0% relative reduction*)<br>time lag = 24 minutes**<br>reduction rate = −5.63 TECU/hr<br>recovery rate = +2.81 TECU/hr |
| GDU<br>eclipse timing:<br>start 03:31 UTC<br>max 05:28 UTC<br>(obsc = 94.1%)<br>end 07:22 UTC | —— N/A —— | —— N/A —— | —— N/A —— | min $\Delta$TEC = −5.01 TECU*<br>reached at 05:51 UTC<br>(28.5% relative reduction*)<br>time lag = 23 minutes**<br>reduction rate = −5.91 TECU/hr<br>recovery rate = +3.99 TECU/hr |
| PTK<br>eclipse timing:<br>start 03:45 UTC<br>max 05:44 UTC<br>(obsc = 93.0%)<br>end 07:32 UTC | 04:50–06:20 UTC<br>(duration: 90 minutes)<br>initial foF2 = 6.21 MHz<br>$(4.6 \times 10^5$ el cm$^{-3}$)<br>foF2 reduction rate<br>= −0.51 MHz/hr<br>(−20.6 el cm$^{-3}$/sec) | min foF2 = 5.44 MHz<br>reached at 06:20 UTC<br>$\Delta$foF2 = −1.58 MHz*<br>(22.4% relative reduction*)<br>time lag = 36 minutes** | 06:20–08:55 UTC<br>(duration: 155 minutes)<br>final foF2 = 6.67 MHz<br>$(5.5 \times 10^5$ el cm$^{-3}$)<br>foF2 recovery rate<br>= +0.48 MHz/hr<br>(+19.9 el cm$^{-3}$/sec) | min $\Delta$TEC = −5.45 TECU*<br>reached at 06:00 UTC<br>(27.9% relative reduction*)<br>time lag = 16 minutes**<br>reduction rate = −5.64 TECU/hr<br>recovery rate = +4.54 TECU/hr |

\* change in ionospheric quantity measured relative to baseline curve (rather than to pre-eclipse initial value)

\*\* time delay since the point of maximum eclipse until the said ionospheric quantity reached minimum