# Peer review of "Ionosonde and GPS Total Electron Content Observations during the 26 December 2019 Annular Solar Eclipse over Indonesia"

_EGUsphere, 2022_

## Author Response (AR1)

**Response to Reviewers' Comments**

We would like to thank the editor and the referees for useful comments and questions on the material presented in the manuscript. Based on the referees' comments, we have made some additions and modifications to the manuscript. In the annotated version of the revised manuscript, these additions and modifications are highlighted in red colors. Below are our detailed item-by-item response to the referees' comments. Here, our response is given in blue and/or red colors.

Response to Referee #1

Major comments:
Introduction:
Kindly mention clearly what is the new findings of this work.

As suggested by the referee, in the revised manuscript we have added an emphasis on new findings from this investigation in the Introduction (lines 68-69 in the revised manuscript).

Figures:
The resolution of some of the labels of some Figures needs to increase, they appear blurred.

In the revision, we have enhanced the resolution of figure labels that appear blurred in the preprint. In particular, we increased the resolution of the labels in Figure 2.

Methodology and Results:
Please improve the methodology. I suggest you give more details on how the TEC was estimated with some equations and references. Similarly, the methodology of the keogram should be elaborated in detail. I would like to encourage the authors not to assume, the readers are already familiar with the techniques.

In the revised manuscript, we have included additional details regarding the TEC calculation, including fundamental equations and related references (lines 109-125). In the revised manuscript, we have also included a more detailed description on the construction of the keograms (lines 151-161).

Specific comments:
Figure 2 if it is possible I will suggest the authors improve the resolution.
For the Figures with subpanels, please label them for easy identification. e.g., Figure 14(a(i)).

In the revised manuscript, we have enhanced the resolution of Figure 2 axis labels. For Figures 12-15 which contain multiple subpanels (in rows and columns), we have now

modified the labeling with roman numerals (i-iv) in addition to the letters (a, b, c) for ease of identification by the readers in general.

Abstract:

**1. Line 10: Kindly rewrite this sentence for clarity.**
This sentence has been modified to improve clarity (lines 9-10 in the revised manuscript).

**2. Line 18: Put "of" before "some".**
We have changed "some" into "of some" (line 18 in the revised manuscript).

Introduction:

**1. Line 36: Change "solar local time" to "local solar time".**
We have changed the wording "solar local time" into "local solar time" (line 35 in the revised manuscript).

**2. Lines 45-48: Please kindly rewrite "For many decades [...]; ...Hairston et al., 2018)**
We have rewritten this sentence in order to make it simpler and easier to understand by the readers (lines 44-47 in the revised manuscript).

Instrumentation and Methodology:

**1. Line 73: Please remove "Relatively".**
We have removed the word "relatively" (line 73 in the revised manuscript).

**2. Lines 92 - 93: Change (x.xx°S yyy.yy°E) and anywhere in the text to (x.xx°S, yyy.yy°E).**
We have modified the format of geographic coordinates in various places in the revised manuscript (highlighted with red color in the annotated version).

**3. Line 128: Kindly rephrase the sentence "Further in the analysis, TEC data detrending was also performed".**
We have rephrased this sentence to make it clearer (line 143 in the revised manuscript).

**4. Lines 128 - 130: Change "Two types of data detrending were performed: one to derive $\Delta$TEC (general deviations from the normal condition) and another to derive TECP (wavelike perturbations with much smaller 130 amplitudes and finer structures)" to "Two types of data detrending were performed: (1) to derive $\Delta$TEC (general deviations from the normal condition) and (2) to derive TECP (wavelike perturbations with much smaller 130 amplitudes and finer structures)".**
We have modified the sentence based on the suggestion given by the referee (lines 143-145 in the revised manuscript).

**5. Lines 133 - 134: Kindly rephrase the sentence "Only after completing the detrending process on the IPPs did we spatially map the TECP values onto fixed grid point(s) for data display."**

We have restructured this sentence in order to improve its clarity (lines 149-150 in the revised manuscript).

Observation Results:

Ionosonde Observations

**1. Lines 144 - 145: Rewrite as "one in the southeast of the solar disk and the other in the northwest of the solar disk."**

We have rewritten this sentence following the given suggestion (line 171 in the revised manuscript).

**2. Lines 188 - 190: Please rephrase as: The recovery phase occurred over a duration of 155 minutes, starting at 06:20 UTC (13:20 LT) until 08:55 UTC (15:55 LT) with an increase in foF2 by 1.23 MHz (from 5.44 MHz to 6.67 MHz).**

We have rephrased this sentence based on the suggested sentence structure (line 219-221 in the revised manuscript).

**3. Lines 195 - 196: change "…while that over Pontianak was 83 minutes" to "…whereas that over Pontianak was 83 minutes".**

We have modified the sentence following the suggested wording (lines 225-227 in the revised manuscript).

**4. Line 235:  "climb" to "ascent"**
**5. Line 242:  "climb" to "ascent"**

We have modified the wording of "climb" into "ascent" (lines 266 and 273 in the revised manuscript).

GPS TEC Observations

**1. I will suggest the authors change "keogram plot" to "keogram".**

We have changed the phrase "keogram plot" into "keogram" in various places in the revised manuscript.

**2. Line 293: Kindly change "…since at this time of day, […] maximum level." to "…since at this time of the day, […] maximum level."**

We have rephrased this sentence following the given suggestion (line 324 in the revised manuscript).

**3. Line 303 - 304: Please rephrase this sentence - "Not until nearing the maximum eclipse did ΔTEC started to drop, which eventually reached approximately -6 TECU at its lowest".**
We have rephrased this sentence in order to make it simpler and easier to understand by the readers (lines 334-335 in the revised manuscript).

**4. Line 342 -343: Change to: "The further away the striped patterns were from the alignment with the C1/max/C4 epoch lines, the more likely they are to be associated with AGW/TID."**
We have modified this sentence following the given suggestion (lines 373-374 in the revised manuscript).

Solar EUV Illumination Variability

**1. Line 370: Please change "Further," to "Furthermore,"**
We have changed this following the suggestion (line 401 in the revised manuscript).

 Discussion

**1. Line 404: I suggest you change "97 minutes and 83 minutes." to "97 and 83 minutes."**
**3. Line 417: Same as comment #1, line 404.**
We have modified these phrases following the suggestion in order to make them more compact (lines 435 and 456 in the revised manuscript).

**2. Line 408-409:  Please insert "e.g.," in the citation as (e.g., Farges et al., 2001; Adeniyi et al., 2007; Goncharenko et al., 2018).**
We have added "e.g." based on the suggestion (line 440 in the revised manuscript).

**4. Line 438 - 440: This overshoot might have been caused by an inward shift of the EIA crest position during the post-eclipse period, after an outward shift that happened earlier during the eclipse (Aa et al., 2020). Please have done any analysis to prove this point in this study?**
Regarding this overshoot phenomenon during the recovery phase, a more detailed description and chain of logic are as follows.  In a past research work reported by Aa et al. (2020) <https://doi.org/10.1029/2020JA028296> on the same solar eclipse event, there was an outward shift of the EIA crest during the eclipse based on TEC and satellite measurement data.  This outward shift was explained by Aa et al. (2020) in terms of enhanced eastward polarization electric field during the eclipse which strengthened the equatorial fountain, making the fountain flow landed over the greater |MLAT| locations.  In the analysis of our processed TEC data, we confirmed that the maximum TEC in the EIA zone during the eclipse was located farther away from the geomagnetic equator line, compared to pre-eclipse. Further, we also found that several hours after the eclipse had ended, the EIA crest configuration was back to normal, just like EIA configuration on regular days.  This is

consistent with the fountain returning to normal strength as the enhanced polarization electric field diminished.  Thus, we conclude that the calming of the fountain effect must have acted as a restoring process to reverse the outward shift of the EIA crest from earlier stage, which would mean an inward shift of the EIA crest as part of the recovery toward the end of the eclipse.

In the revised manuscript, we have included an additional discussion (lines 479-485) in order to elaborate this point.

**5. Line 469: Please put a colon after "includes" as … includes: (1) ….**
We have modified this following the suggestion (line 512 in the revised manuscript).

Response to Referee #2

Line 121: I suggest including information about elevation angles used in TEC calculation.
In the TEC calculation, the cutoff for the elevation angle was 20 degrees.  In the revised manuscript, we have now included this additional information (line 135).

Line 192: Correct the word "ecipse" to eclipse.
The typographical error has been fixed (i.e. "eclipse") in the revised manuscript (line 223).

Figures 5-7: It is clear that IRI does not represent the ionospheric characteristics of Indonesia.  I suggest using the ionospheric average from the ionosonde instead of the IRI model.
Regarding the baseline curves in Figures 5-7, we followed the suggestions given by the referee.  In the revised manuscript, we now use the averages of ionosonde observations to form the baseline curves.  The absolute and percentage values of foF2 and foF1 reductions also changed slightly as a result of this baseline modification, although they do not alter the overall conclusion of the investigation.  These changes are highlighted with red color in various places of the revised manuscript.

As the referee remarked, the performance of the IRI model over the Southeast Asian region, more specifically over the Indonesian region, turned out to be less than optimal.  Detailed quantification on the IRI model performance over this geographic region may also be investigated further in future research.

Line 340: What are the acoustic gravity wave characteristics observed during the eclipse?  The problem is the authors did not comment on the origin and generation of TIDs during the eclipse. Therefore, it is out of scope and I suggest removing AGW from the text.
Following the suggestion, here we skip the mention of AGW due to limited information on the wave parameters.

Line 349: Why is the Laplacian the better approach to capture the inhomogeneity? The authors have to give to read more details about the technic and explain the vantages and disadvantages of this technic to study these differences in data.

The main advantage of the Laplacian operator for capturing inhomogeneity in the form of sharp discontinuity in 2-D data is its low computational cost.  In addition, the Laplacian operator has the same properties in each direction (i.e. isotropic), which simplifies the interpretation of results.  Unfortunately, the edge direction is unavailable, which is its disadvantage.  Nevertheless, edge direction is not very important in our situation, and our analysis was not impacted.

In the revised manuscript, we have included more detailed explanation and foundational references regarding the use of Laplacian technique (lines 192-197).

Line 407: I suggest discussing the present results with results observed over the South American sector. See Resende et al. (2021) https://doi.org/10.5194/angeo-2021-61

In the revised manuscript, we have included some additional discussion (lines 447-454) of the present results in relation to recent results reported by Resende et al. (2022).

Line 482: Which is the physical mechanism that may explain these phenomena?

Regarding the observed "shallow TEC valley" pattern, at this stage we do not know the precise physical mechanism that may have caused the phenomena.  What we have done was to eliminate as much as possible various scenarios involving instrumental artefact and geographical distribution of ground-based observing stations, which could conceivably lead to such a "shallow TEC valley".  One scenario under consideration was a systematic and correlated shift in the TEC bias for a group of nearby receiver stations.  However, each receiver device is operating independently (even when their spatial distances are quite close), which makes it unlikely for their hardware biases to be linked electronically.  Another scenario under consideration was slant factors that may be quite extreme (due to low elevation angles) for IPPs that are located over ocean region unpopulated by receiver stations.  However, it turned out that this latter scenario predicts that the "shallow TEC valley" would instead have happened over the ocean region (opposite to the observed fact that the "shallow TEC valley" actually occurred over land mass region populated with receiver stations).  Hence, we can rule out this possibility as well.  Therefore, the exact physical mechanism for the occurrence of this "shallow TEC valley" is still an open scientific question for the community.  Nevertheless, some major possible instrumental effects have been ruled out in our considerations.

In the revised manuscript (lines 524-534), we have expanded the discussion on this point.

---

## Referee Report (RR1)

The authors present evidence of the effect of the 26 December 2019 solar eclipse on the ionosphere. They showed this using Total Electron Content (TEC) data from Global Navigation Satellite System (GNSS) receivers over the Indonesian region. Also, ionosonde data from Canadian Advanced Digital Ionosonde (CADI) in two locations were used to complement the TEC observation. Using the Solar Dynamics Observatory (SDO), they tracked the umbra of the eclipse spatiotemporally. Their investigation methods are clear. The authors gave much emphasis to the effect of the eclipse on the ionosphere particularly the reduction in the TEC and Ionosonde observations as well as the time delay. This current work contributes to literature by showing how this type of eclipse affects the ionosphere in the Indonesian region. The authors addressed all the comments pointed out in the previous version of the manuscript. The current version of the manuscript has improved. I, therefore, recommend the work be published after the implementation of the comments and corrections.

**Comments:**

**Abstract**

**Page 1, Line 1: Change to -** We report the investigation of the ionospheric response to the passage of an annular solar eclipse over Southeast Asia on 26 December 2019, using multiple sets of observations.

**Page 1, Lines 14-15: Change to -** The GPS TEC data mapping along a set of cross-sectional cuts indicate that the greatest TEC reduction actually occurred to the north of the solar eclipse path, opposite of the direction from which the lunar shadow fell.

**Introduction**

**Page 2, Line 30: Change to -** ….. (see Aplin et al. (2016) for an overview).

**Page 3, Lines 54-55: Change to** ………………. ionospheric response to solar eclipse ……….
. I think "ionospheric response" is more suitable in this context. This is because here, you are investigating the effects of the solar eclipse on the ionosphere, not the other way round.

**Page 3, Lines 69-71: Rewrite as:** - Section 2 of this paper describes the methodology, Section 3 describes the results, Section 4 presents the discussion of the findings, and Section 5 the conclusion presents.

**2 Instruments and Methodology**

In the instrumentation and methodology, the symbols of the instruments are not defined in the text. I suggest the authors describe the symbols in the text. This should also be done for symbols in the entire manuscript.

**Page 4, Line 112:** - Insert "the" as: From two the frequencies $f_1$ and $f_2$, …………………………….

**Figure 3 Caption, Page 5 and Line 139-140:** The authors only described the path and instantaneous position of the lunar shadow on 26 December 2019 (panel b) but did not do that for the other days (i.e. panels (a) and (c)). I suggest you give these details as well.

**Page 6, Lines 151 – 161:** I will suggest the authors present a sample keogram.

**3 Observation Results**

**Page 10, Line 285:** - Change "and" to "to" as: The baseline TEC was determined by averaging the TEC values from 25 to 27 December 2019, ………..

**Page 11, Lines 321-322:** - On the left panel of Figure 12a, ………… I think, instead of using the left panel, the authors should use the sub-labels, that is, (i), (ii), and (iii). So, the labeling can be Figure 12a(i). This will make it easier for the reader to follow. Please check through the entire text and make these changes.

**Page 11, Line 334:** - Please change ……….. nearing the maximum eclipse. ……. to ……….. approaching the maximum eclipse.

**Page 12, Line 367:** - The tip of this bulge occurs started at latitude … . Please remove the "occurs".

**Page 13, Line 390:** - Please, there is a repetition of "the", kindly remove one.

**4 Discussion**

**Page 15, Lines 465-466: -** Please change "The significant difference with Adeniyi et al.'s (2007) findings might be due to the different type of solar eclipse that occurred" to "The significant difference in the findings of Adeniyi et al. (2007) might be due to the different type of solar eclipse that occurred".

**Page 15, Lines 532-534: -** I will suggest that the authors revise this sentence.

---

## Author Response (AR2)

**Response to Reviewers' Comments**

We would like to thank the editor and the referees for useful comments and suggestions. We have made the suggested changes/corrections to the manuscript. Below is the summary.

In the Abstract, we have modified line 1 into "...the investigation of the ionospheric response to..." and lines 14-15 into "...cross-sectional cuts..."

In the Introduction, we have modified line 30 into "see Aplin et al., 2016 for an overview" lines 54-55 into "...ionospheric response to..." and lines 69-71 into "...Section 3 describes the results, Section 4 presents the discussion of the findings, and Section 5 the conclusion."

In the Instruments and Methodology section, we have modified line 112 into "...the two frequencies..."; and we have also modified Figure 3 to include the solar eclipse path line visualization on 25 and 27 December 2019, plus a sample TEC keogram in panel (d).

In the Observation Results section, we have made the suggested changes:
In line 285, we have modified the sentence into "...averaging the TEC values on 25 and 27 December 2019..." (26 December 2019 was excluded from the baseline TEC averaging)
In lines 321-322, we have modified the labeling to make use of the sub-labels.
In line 334, we have modified the wording into "...approaching the maximum eclipse..."
In line 367, we have removed the word "occurs"
In line 390, we have removed the extra "the"

In the Discussion session, we have modified lines 465-466 into "...The significant difference in the findings of Adeniyi et al. (2007)..." and we have also revised lines 532-534 into "This scenario predicts that the 'shallow TEC valley' would have happened over the ocean region. However, the 'shallow TEC valley' was in fact found over land mass (well-populated with receiver stations), contrary to what this scenario predicts. Hence, we can rule out systematic bias shift and extreme slant factor as the root cause of the 'shallow TEC valley' feature. As such, a few major instrumental artefacts have been ruled out, and the question regarding the responsible physical mechanism remains open."